# Medicinal Chemistry of Anti-HIV-1 Latency Chemotherapeutics: Biotargets, Binding Modes and Structure-Activity Relationship Investigation

**DOI:** 10.3390/molecules28010003

**Published:** 2022-12-20

**Authors:** Yan-Kai Wang, Long Wei, Wei Hu, Pei-Xia Yu, Zhong Li, Hai-Peng Yu, Xun Li

**Affiliations:** 1School of Pharmacy and Pharmaceutical Sciences & Institute of Materia Medica, Shandong First Medical University & Shandong Academy of Medical Sciences, NHC Key Laboratory of Biotechnology Drugs (Shandong Academy of Medical Sciences), Key Laboratory for Rare & Uncommon Disease of Shandong Province, No 6699, Qingdao Road, Ji’nan 250117, China; Key Laboratory of Forensic Toxicology, Ministry of Public Security, Beijing 100192, China; 2Shandong University, No 72, Binhai Road, Qingdao 266237, China

**Keywords:** HIV-1, latent reservoirs (LVRs), latency-reversing agents (LRAs), resting CD4^+^ T cells (rCD4s), anti-HIV-1-latency

## Abstract

The existence of latent viral reservoirs (LVRs), also called latent cells, has long been an acknowledged stubborn hurdle for effective treatment of HIV-1/AIDS. This stable and heterogeneous reservoir, which mainly exists in resting memory CD4^+^ T cells, is not only resistant to highly active antiretroviral therapy (HAART) but cannot be detected by the immune system, leading to rapid drug resistance and viral rebound once antiviral treatment is interrupted. Accordingly, various functional cure strategies have been proposed to combat this barrier, among which one of the widely accepted and utilized protocols is the so-called ‘shock-and-kill’ regimen. The protocol begins with latency-reversing agents (LRAs), either alone or in combination, to reactivate the latent HIV-1 proviruses, then eliminates them by viral cytopathic mechanisms (e.g., currently available antiviral drugs) or by the immune killing function of the immune system (e.g., NK and CD8+ T cells). In this review, we focuse on the currently explored small molecular LRAs, with emphasis on their mechanism-directed drug targets, binding modes and structure-relationship activity (SAR) profiles, aiming to provide safer and more effective remedies for treating HIV-1 infection.

## 1. Introduction

Acquired immune deficiency syndrome (AIDS) caused by human immunodeficiency virus-1 (HIV-1) infection remains an incurable disease largely due to the existence of a persistent latent reservoir, which has been the last bastion for effective treatment of HIV-1/AIDS. In 1995, Siliciano and co-workers identified for the first time the latent cells in memory resting CD4^+^ T cells (rCD4s) from HIV-1-infected patients and proposed a term “latent reservoir” to graphically depict an ingenious survival method of the HIV-1 provirus [1]. In addition to rCD4s, the latent cells also exist in monocytes, macrophages, lymphocytes, glial cells, astrocytes, natural killer cells, and multi-functional hematopoietic stem cells. Actually, humans have some immune exemption sites that offer a physiological tissue condition for latency of HIV virus, covering various lymphoid organs and tissues, including but not limited to the spleen, lymph nodes, abdominal and intestinal related lymphoid tissues, central nervous system, testis and other reproductive systems. In this regard, LVR represents a common term for all cells, tissues or any anatomical sites where a virus hides [2].

LVR is a complicated and heterogeneous phenomenon, involving multiple and interlinked factors both at transcriptional and post-transcriptional levels. Accordingly, HIV-1 latency can be divided into two categories: pre-integration latency and post-integration latency. When transcriptional activators are lacking or chromatin structure is concentrated, latency can be maintained at a transcriptional level, while when nuclear RNA transportation or microRNA translation is inhibited, latency is usually maintained at the post-transcriptional level [3].

Currently, the underlying mechanisms of HIV-1 latency are incompletely understood. At least six molecular mechanisms for illustrating the establishment and maintenance of LVRs have been proposed, as displayed in Figure 1: (i) epigenetic regulation of viral gene expression, e.g., methylation, acetylation, deacetylation, phosphorylation and ubiquitination at histone tails; (ii) accessibility of activation-dependent cellular transcription factors, such as host transcription factors nuclear factor κB (NF-κB), nuclear factor of activated T cells (NFAT), activator protein 1 (AP-1) and positive transcriptional elongation factor b (P-TEFb), etc., which are crucial factors for active HIV transcription; (iii) influence of proviral integration site; (iv) influence of microRNAs (miRNAs) on viral gene transcription; (v) RNA elongation, splicing and transport; and (vi) formation of antisense HIV genomic transcript, which will down-regulate gene expression. In some cases, the aforementioned factors play synergistic and/or antagonistic effects on HIV latency. However, in view of the invisibility and complexity of LVRs, the exact mechanism about why and how HIV virus preferentially establishes latent infections in rCD4s is still not understood, which leaves the efforts to eliminate the latently infected cells unsuccessful to date [4].

Much evidence from both animal and human models have shown that LVRs have been formed within days of HIV-1 infection by directly infecting rCD4s, or initially infecting activated CD4^+^ T cells, which then convert into a quiescent state. These stable LVRs that harbor an integrated but replication-competent proviruses can remain for a long time in the body, with an estimated half-life as long as 40~44 months. In other words, HIV virus might hole up in host cells for up to 73 years without triggering obvious symptoms, which renders HIV/AIDS an almost incurable disease [5]. Besides, LVRs are extremely difficult to eradicate or reduce, because they are transcriptionally silenced by expressing little or no viral proteins, making them not only readily resistant to combination antiretroviral therapies (cARTs) but difficult to detect and purge for the host immune system. However, cells in LVRs can reactivate at any time and produce more viruses, leading to rapid viral rebound once antiviral treatment is interrupted [6]. Therefore, the exploitation of an effective and safe anti-HIV-1-latency remedy remains a highly prioritized goal.

To date, multiple eradication interventions against HIV-1 reservoirs have been put forward, with the expectation of achieving a functional cure for HIV-1/AIDS. Currently, the broadly accepted regimen to combat the LVRs chiefly relies on a ‘shock and kill’ approach, which involves a two-step protocol. First, drugs called latency-reversing agents (LRAs) are utilized to reactivate hiding viruses by stimulating viral protein expression in rCD4s (‘shock’). Second, the reactivated cells, which are now susceptible to cytolytic T lymphocytes or virus-induced cytopathogenicity, can then be readily eliminated by cARTs together with host immune-mediated interventions, or other therapeutic regimens (‘kill’). To ensure the successful implementation of this strategy, the identification of effective and safe LRAs is a prerequisite [7].

Global T cell activators were initially developed to function as LRAs to reactivate proviruses in rCD4s, but severe toxicities (e.g., heart attack and temporary blindness) impelled researchers to seek safer LRAs that merely induce HIV-1 provirus expression without affecting normal immune functions in the body. Benefiting from multiple in vitro and in vivo HIV-1 latently infected models, different sorts of small molecular LRAs with distinct molecular mechanisms have been developed in succession for this purpose, as diagrammed in Figure 1. Among these identified LRAs, three major types of LRAs with different mechanisms are expanding research hotspots. Molecules of the first type regulate epigenetics and include histone deacetylase inhibitors (HDACIs), DNA methyltransferase inhibitors (DMTIs) and histone methyltransferase inhibitors (HMTIs). Molecules of the second type activate transcriptional factors (e.g., NF-κB and AP1) and include protein kinase C (PKC) activators, CCR5 antagonists, and STAT5 agonists. Molecules of the third type mainly refer to bromodomain and extra-terminal domain inhibitors (BETIs), exerting functions by promoting transcription elongation [8].

So far, albeit with positive progress in developing diverse chemotypes of LRAs with underlying biotargets and mechanisms of action, there has not yet been a significant breakthrough in successfully eliminating the latent proviruses. Most of these LRAs are ineffective in reducing the overall size of LPRs. Thus, novel LRAs with better therapeutic efficacy and lower toxicity are still urgently desirable. In the following subsections, we mainly focus on the “shock” aspect, with the emphasis on the description of potential drug targets, binding patterns as well as SAR perspectives of corresponding LRAs that are relevant to anti-HIV-latency chemotherapies.

## 2. Possible Biotargets and Related LRAs

### 2.1. Histone Deacetylase Inhibitors (HDACIs)

HDACs are a group of epigenetic enzymes that can significantly affect chromatin topology and the histone deacetylation process by removing functional acetyl groups from the N-terminus of lysine residues and facilitating a high-affinity interaction between histones and DNA backbone, leading to a condensed inactive chromosomal DNA structure and consequent blockage of gene transcription. There are a total of 18 isoforms of mammalian HDACs, which are divided into four classes (class I, II, III and IV), largely based on the sequence homology, cellular location and folding mode of peptide chains. Class I contains three subtypes, IA (HDAC1, HDAC2), IB (HDAC3) and IC (HDAC8). Class II includes two subtypes, class IIa and IIb, in which class IIa consists of HDAC4, HDAC5, HDAC7 and HDAC9, while class IIb includes HDAC6 and HDAC10. Class III HDACs, also known as sirtuins, are nicotinamide adenine nucleotide (NAD)^+^-dependent enzymes, which contain seven members (sirtuins 1~7). Class IV contains only HDAC11. HDACs generally refer to zinc-dependent class I, class II and class IV isozymes, unless noted otherwise [9].

HDACs contribute to proviral gene silencing of HIV latency by directly deacetylating histones at proviral integration sites (5′-long terminal repeat, 5′-LTR) or by indirectly inducing deacetylation of non-histone proteins (e.g., NF-κB). Thus, inhibiting HDACs can facilitate changes in chromatin architecture and recruitment of host transcription factors to LTR, leading to the acceleration of viral transcription. Using small molecular HDACIs as epigenetic modifiers thereby represents a viable and predominant strategy to eliminate latent reservoirs [10].

Structurally, currently identified HDACIs typically follow a common pharmacophoric feature by mimicking the structure of the natural substrate lysine, which comprises a surface recognition cap moiety that can tolerate structural variability to accommodate the broad hydrophobic region of HDAC (Cap region); a functional zinc-binding group (ZBG) that can orient and coordinate with the catalytic zinc ion; and a linear or cyclic linker with 5~7 atoms (Linker) that traverses the long and narrow tunnel to connect the Cap portion and ZBG [11].

In the past decades, a variety of HDACIs hits or candidates that vary in skeletal structures have been developed by modulating these pharmacophoric fragments and submitting to bioevaluation in various stages, aiming at achieving both elevated potency and isoform selectivity against HDACs-mediated pathological conditions, particularly hematological malignancies and solid malignant tumors. At present, vorinostat (SAHA), which has been approved for the treatment of cutaneous/peripheral T-cell lymphoma, is by far the clinically best-studied HDACI to be fully assessed for latency-related anti-HIV-1 therapies, either as a single regimen or in combination with other types of LRAs. SAHA showed promising in vitro HIV-1 latency-reversing effects in multiple HIV-1 latently infected cell lines (e.g., ACH2, U1 and J-Lat) and a latent provirus that was isolated from resting CD4^+^ T cells in HAART patients. However, the therapeutic outcome from pan-HDACI SAHA is widely limited by its insufficient selectivity towards specific isoforms, resulting in many unwanted side effects, including dehydration, anorexia, thrombocytopenia, arrhythmia, and also poor pharmacokinetic (PK) profiles. Hence, intensive structural modifications have been carried out to procure more potent HDACIs with improved selectivity and less toxicity [12].

To achieve improved efficacy towards HIV-1 latent reservoirs, Okamoto et al. presented a spectrum of structural mimics of SAHA by employing a structure-based drug design protocol, based on the obtained crystal structure of HDAC in complex with SAHA. Specifically, the hydroxamic acid of SAHA that acts as a ZBG was initially replaced by an acylated thiol group, while the cap moiety was prolonged by inserting a thiazole or phenyl motif. It resulted in HDAC inhibitory potencies of two compounds, NCH-51 (**1**) and NCH-51 (**2**), increasing by about 2–3 fold over that of prototype SAHA, with selective preferences towards HDAC1/4/6 isoforms. Additionally, compared with congener 1, NCH-51 has more potent HIV latency-reversing effects in latently infected cell lines, OM10.1 (CC_50_ = 2.2 µM) and ACH-2 (CC_50_ = 2.4 µM), without causing obvious cell death, making it a promising lead. Encouragingly, owing to the introduction of an acylated thiol group instead of hydroxamic acid, which is supposed to have a potential mutagenicity side effect, these two compounds exhibited better PK profiles and lower cytotoxicity than the parent compound SAHA [13].

Considering that the phosphate group can also function as a potential ZBG, Etzkorn et al. hypothesized that the substitution of phosphate analogues (phosphonamidate, phosphonate, phosphinate) for hydroxamic acid might afford equal potency while avoiding the limitations of hydroxamic acid. Keeping this in mind, they retained the cap part of SAHA unchanged and modified the linear linkage with a one-carbon degradation. Unfortunately, the resulting SAHA analogues **3**~**5** lost HDAC-binding affinities. One possible reason might be that the negatively charged phosphates may sterically hinder the coordinative contact with the catalytic zinc ion of HDACs, resulting in dramatically impaired efficacies [14].

To discover more SAHA-derived HDACIs, Pflum et al. focused on the bridged linkers to explore chemical diversity and enrich SAR information. To this end, a series of SAHA analogues **6**~**9** were synthesized by appending branched substituents with different sizes and dispositions. Surprisingly, except for the benzyl-attached compound **8** that only gave a decent HDAC8 inhibitory affinity, nearly all these compounds furnished significant lowered potencies but exerted preferential selectivity towards HDAC6/8 versus HDAC1/2/3 compared with the archetype SAHA, indicating that structural modifications on bridge portion might be conducive to selective inhibition for HDAC6/8 [15].

As the phosphorus-based ZBGs might largely account for the unfavourable HDAC inhibition, further structural optimizations were then focused on other parts of SAHA. Apicidin, a fungal metabolite bearing a cyclic tetrapeptide unit, is a naturally originated HDACI (IC_50_ = 0.7 nM) [16]. Inspired by this, Etzkorn and co-workers postulated that the macrocyclic peptide portion might act as a hydrophobic surface recognition group (Cap), and further modifications were conducted by simplifying the macrocyclic skeleton and constructing a linear chain of five carbon atoms (linker). Considering the strong zinc-binding affinity, the ethyl ketone part of Apicidin was replaced with hydroxamic acid (ZBG). As expected, the yielded hybrid 10 gave a 2-fold improvement in HDAC inhibition and nearly a 5-fold selectivity for HDAC1 over HDAC8. To pursue this lead further, a ring-opening operation was implemented from different locations to expose carboxylic acid and aryl substituent, respectively, aiming to examine the influences of rigid macrocycles and flexibility on activity and selectivity. The resulting compound **11**, however, decreased both activity and selectivity, while another ring-opening product, **12**, gave comparable efficacy only with its precursor, 10, implying that the hydrophobicity and conformational distribution of the cap region may have a significant impact on HDACs inhibition and selectivity, and the restricted macrocycllic ring is more suitable than the flexible side chains for HDAC1-selective inhibition. Likewise, the impaired potency of compound **11** in comparison with **12** might also be attributed to the charged groups [17].

The aforementioned efforts on the structural variations derived from SAHA or Apicidin, as seen in Figure 2, have provided useful SAR information for further identification of more potent HDACIs, although these newly identified compounds **3**~**12** were not submitted to the HIV-1 reactivation evaluation. Actually, similar to SAHA, in the past decades many synthetic and naturally available HDACIs as anticancer agents, including but not limited to the hydroxamic acid-based givinostat (ITF2357), panobinostat (LBH589), nanatinostat (CHR-3996), pracinostat (SB939) and belinostat (PXD101); benzamide-based mocetinostat, entinostat (MS-275), as evinced in Figure 3, also displayed favorable HIV latency-reversing activities in various latently infected cell lines, and these agents are typically well-tolerated by participants. Among them, thiol-based romidepsin is the most effective LRA to date, which has supported the “proof-of-principle” that latent reservoirs can be safely activated, and perhaps, entirely eliminated [18]. Rasmussen et al. compared the effects on HIV production in latently infected cells (U1 and ACH2) as well as T-cell activation of several hydroxamic acid-based HDACIs that were undergoing clinical development. The results indicated that these HDACIs gave different degrees of HIV-1 reactivation potencies at therapeutic concentrations, with activity order of panobinostat > givinostat ≈ belinostat > SAHA. However, all these HDACIs induced moderate T-cell activation, which hindered their further clinical application [19].

As we know, HDACI treatment in HIV-1-infected individuals generally suffers from abnormal T-cell activation and nonspecific HDAC inhibition, leading to undesired HIV-1 persistence and other side effects by causing clonal expansion of latently infected rCD4s, which has been the major bottleneck for warranting further clinical investigation [20]. Thus, an ideal LRA candidate for achieving the desired ‘shock and kill’ tactics should have the capability of stimulating latent HIV-1 transcription without provoking homeostatic proliferation and/or extensive T-cell activation that is highly correlated with the cytokine release, so as to avoid possible immune hyperactivation and the consequent concomitant cytokine storm, also known as cytokine release syndrome (CRS), as well as acceptable PK properties [21]. Fimepinostat (CUDC-907), a dual inhibitor of class I-selective HDACI and PI3Kα, might be a suitable LRA candidate. It not only displayed comparable latency-reversal activity with romidepsin, the current most effective HDACI tested in anti-HIV-1 trails, at the cellular level, but caused reduced T-cell activation without any negative influence on T cell proliferation [22].

Accumulating evidence indicates that class I-selective HDACIs—especially HDAC-1, -2 and -3, with HDAC3 isoform being the most important, functioning as transcriptional “on switches” of latent viruses and maintaining the deacetylated state of reactivation-related transcription factor nuclear factor κB (NF-κB)—might be more effective than pan-HDACIs in eradicating HIV latency by inducing more latent proviruses [23]. Taking thiol-based orally active pan-HDACI ST7612AA1 (**13**), as another example, acting as a prodrug and potent HIV latency activator, ST7612AA1 actually exerts an HIV reactivation effect by transforming into its active form, ST7464AA1 (**14**), a class I-selective HDACI, via in vivo hydrolysis [24]. For instance, Lewin SR and co-workers compared the HIV-1 reactivation efficacies of class I-targeted HDACI (entinostat) and three pan-HDACIs, SAHA, panobinostat and oxamflatin (metacept-3, MCT-3); their results also proved that entinostat gave the most potent HIV-1 latency-reversal activity by inducing more viral expression [25]. The fact that HDAC3 highly selective inhibitor BRD3308 not only was active for latent HIV-1 reactivation in 2D10 cell model but could induce viral outgrowth from rCD4s of antiretroviral-treated patients further proved that class I-targeted HDACIs, especially HDAC3 inhibitors, are particularly effective anti-latencyt agents with improved HIV-1 reactivation potencies and fewer effects on other unrelated cellular genes [26].

Similarly, to discover more potent class I (HDAC1/2/3)-selective HDACIs for eliminating a latent reservoir, Yu and co-workers from Merck have develop an array of ethyl ketone-based macrocyclic HDACIs, referring to the skeleton of class I-selective HDACI apicidin via ring expansion while retaining the linker and ZBG (ethyl ketone) unchanged, since the macrocycle structure is believed to have greater binding affinity with HDAC2 subtype, as evinced in Figure 4. The facts that the two most potent class I-selective macrocyclic HDACIs, **15a** and **15b**, gave both enhanced class I HDAC inhibition and HIV latency-reversal potencies, supported the strategy for designing macrocyclic class I-selective HDACIs as promising LRAs, to a great extent. However, due to relatively low bioavailability of macrocycles, the PK profiles of these macrocyclic HDACIs still need further improvement [27].

Granted, there are some cases that have proven otherwise. For example, another two effective class I-selective HDACIs, nanatinostat and romidepsin, as shown in Figure 3, could not induce the generation of viral antigens or particles from rCD4s, partially owing to the lack of effective accumulation of spliced viral transcripts, despite in vitro effectiveness in latency-infected cell lines. Along with this, both of them impaired the function of CD8^+^ T cells, with romidepsin causing more impairment, which might explain, to a certain degree, the unsatisfactory clinical evaluation of various HDACIs in ARV-suppressed individuals so far [28].

Colletively, as we can see from the above description, although HDACIs, especially class I-HDAC selective inhibitors, are a class of well-acknowledged LRAs with promising in vitro and ex vivo anti-latency effectiveness, which HDAC subtypes actually have a direct impact on HIV-1 latency in vivo is still not yet entirely elucidated. Hence, better investigative tools, whether more potent HDACI-based LRAs or latency screening models, particularly appropriate in vivo evaluation models, that can translate this knowledge into clinical chemotherapies are in urgent demand.

### 2.2. DNA Methyltransferase Inhibitors (DMTIs)

As one of the chief epigenetic modifications, DNA methylation is supposed to play a role in governing HIV latency by epigenetic regulating of 5′-LTR cytosine-phosphate-guanine (CpG) methylation and inhibiting HIV viral transcription initiation. LTR contains two CpG islands and a HIV-1 promoter can be hypermethylated at these two CpG islands, particularly island 2, surrounding HIV transcription initiation sites. Besides, methyl-CpG binding domain protein 2 (MBD2) can specifically bind to methylated DNA and then recruit HDAC1/2, leading to chromatin unwinding and gene silencing and consequently histone deacetylation [29].

The DMTI 5-aza-2′-deoxycytidine (decitabine, 5-aza-CdR, **16**) which has been approved by FDA for the therapy of myelodysplastic syndrome (MDS), as shown in Figure 5, has proved to demonstrate weak HIV-1 reactivation activity but displays an intensified promoting effect when utilized in combination with other HIV-activating agents, such as tumor necrosis factor α (TNFα), PKC activator prostratin [30] or HDACIs in most J-Lat cell lines [31]. However, the HIV-1 reactivation ability of decitabine, either used alone or in combination with other types of LRAs, displayed strong cell type dependence, implying a more comprehensive assessment should be carried out when using decitabine as a LRA in HIV reactivation trials. Another DMTI azacitidine (5-azacytidine, **17**) also could induce latent HIV-1 proviruses [32].

Lint and co-workers have observed that decitabine-induced HIV-1 reactivation was accompanied by a decreased recruitment of ubiquitin-like with an epigenetic integrator, PHD and RING finger domain 1 (UHRF1), to the viral promoter, implying UHRF1 might be a promising pharmacological target for discovering potent LRAs. To provide a demonstration of this proof-of-concept finding, this lab further utilized epigallocatechin-3-gallate (EGCG, **18**), a polyphenol from green tea, which has been reported to possess certain UHRF1 inhibitory activity [33], and NSC232003 (**19**), a specific UHRF1 inhibitor, as two chemical probes, to ascertain the LRA potential of UHRF1 inhibitors. The results exhibited that EGCG partially reactivates latent proviruses through the inhibition of UHRF1, whereas NSC232003 provoked significant HIV-1 reactivation, highlighting the tight correlation of anti-UHRF1 with latency-reversal potency [34].

Despite some positive latency-reversal outcomes from decitabine and azacitidine, the role of DNA methylation in the formation and maintenance of HIV-1 latency, however, is still in dispute. Blazkova et al. detected very low levels of methylated CpG in HIV-1 infected individuals under antiretroviral treatment, probably because activation is limited due to proviral DNA hypermethylation, highlighting the necessity for a deep understanding of the underlying heterogeneity of DNA methylation on HIV-1 latency as well as a more reasonable assessment of DMTIs as latency activators [35].

All in all, since DNA methylation affects both the viral and host genome, silencing of HIV transcription via inhibition of DNA methylation also causes aberrant methylation of the host genome. This process is irreversible, and vice versa, implying that DNA methyltransferase is likely not the ideal strategy for developing effective HIV-1 reactivating agents. However, the DNA methyltransferase-associated UHRF1 is expected to be an attractive pharmacological target to explore more effective LRAs.

### 2.3. Histone Methyltransferase Inhibitors (HMTIs)

In addition to DNA/CpG methylation, histone methylation—which primarily occurs at the *N*-terminal arginine or lysine of H3 and H4 histones and is covalently modified by an epigenetic enzyme, histone methyltransferase (HMT)—is also essential for the establishment of the silencing of HIV-1 transcription and maintenance of genome stability. Four important lysine methyl transferases (KMTs), EZH2, SUV39H1, G9a and G9a-like protein (GLP), are the most extensively studied HMTs.

EZH2 is located at the promoter of latent HIV-1 provirus in T cells, participates in histone H3 lysine 27 trimethylation (H3K27me3), and plays a major role in chromatin-mediated HIV-1 transcriptional regulation and viral suppression. SUV39H1 primarily participates in H3K9 trimethylation (H3K9me3). G9a, also known as euchromatic histone-lysine N-methyltransferase 2 (EHMT2), is responsible for H3K9 dimethylation (H3K9me2) by catalyzing the addition of a methyl group from S-adenosyl-L-methionine (SAM) to a histone lysine residue. G9a-like protein (GLP), also called EHMT1, is another H3K9 methyltransferase with 80% sequence homology to G9a in the suppressor of variegation 3–9, enhancer of zeste and trithorax (SET) domains [36]. Both G9a and GLP, together with SUV39H1, play pivotal roles in transcriptional silencing in HIV-1 latency. Histone methylation marks H3K9me2, H3K9me3 and H3K27me3 in particular have been validated to prevent lysine from being acetylated and meanwhile keep the chromatin in a dense state in both latency cell lines and primary CD4^+^ T cell models, and therefore are considered as important histone methylation marks [37].

Quinazoline BIX01294 (**20**), a specific G9a inhibitor, was the first identified HMTI with clear in vitro HIV latency-reversing activity from a high-throughput screening protocol [38]. The subsequent resolved X-ray co-crystallographic structure of BIX01294 bound to HMT (PDB code: 3K5K) as well as a SAR investigational outcome of BIX01294 have provided useful clues for further structure-based drug design and chemical optimizations [39]. As indicated in Figure 6A, HMT contains three active binding grooves: pockets I and II are two functional solvent regions; pocket III, also referred to as a lysine-binding channel, functions as a methyl transfer and transportation channel (Figure 6A). The quinazoline core of BIX01294 occupies the central histone peptide binding cavity, while a benzyl-substituted piperidine moiety and a 1,4-diazepane fragment extend into pocket I and II, respectively, and a 7-methoxy group orients towards pocket III, a long and narrow region (Figure 6B). Besides, 4-NH, two N atoms of piperidine and a diazepane ring form the key H-bonding force network with three acidic amino acids, Asp1078, Asp1083 and Asp1074, which have significantly contributed to its beneficial HMT inhibitory potency.

To further explore the SARs and discover more potent HMTIs, Jin et al. employed several rounds of chemical optimizations on BIX01294. Initially, a 4-amino substituted group (R^1^) of quinazoline skeleton was investigated. Since the 4-NH forms an important H-bonding force with Asp1083 while the benzyl group has extended beyond the active interfaces without contributing any protein-ligand interactions, the influences of piperidine N and the sizes of substituents on the bioactivity were studied by removing the redundant beznyl portion while retaining the 4-NH unchanged. As a result, compound **21** with 1-methylpiperidine moiety gave an 8-fold improvement in G9a binding affinity (IC_50_ = 0.23 μM) compared with the prototype BIX01294 (IC_50_ = 1.9 μM), indicating that piperidine N shows great promise to increase the drug-like profiles of quinazoline derivatives by not only providing multiple molecule-protein interactions but also producing ameliorated water solubility and bioavailability. By contrast, introduction of small-sized groups (e.g., cyclopropyl or isopropyl) results in obviously impaired potency. As far as 2-substituent (R^2^) was concerned, both N-containing 6-membered (e.g., compounds **22** and **23**) and **7**-membered cycloalkanes were tolerated.

When it comes to R^3^ group, although it oriented towards the water-exposed tolerant lysine-binding region (pocket III), it did not fit this narrow space properly, which has offered an underexploited but valuable site for further exploitation. Among the investigated hydrophilic side chains, introduction of *N*,*N*-dimethylbutane (compound **24**) gave a marked improvement on G9a affinity by providing an additional water-mediated contact and thus yielding more favorable adaptability to the lysine-binding channel, elevating affinity by about 127 times, in contrast to the prototype BIX01294.

The molecular modeling docking result of compound **24** in complex with G9a (PDB: 3K5K), as evinced in Figure 7, has explained, in a large part, why both the multi-site contacts and *N*-bearing side chain that inserts into lysine-binding channel greatly account for its high potency. Specifically, in addition to the hydrogen bond formed between 4-NH and ASP1083, the piperazine N atom forms a salt bridge with ASP1074 and an electrostatic interaction with ASP1078. Besides, 1-N of quinazoline ring is supposed to be protonated under physiological pH conditions, forming a salt bridge with ASP1088. More importantly, the protonated N atom of the *N*,*N*-dimethylbutane part forms not only a hydrogen bond with Leu1086, but also a cation-π interaction with Tyr1154 (displayed as red circle), which contributed much to the binding affinity [40].

Fuchter et al. examined the influences of a central quinazoline ring and 6,7-dimethoxyl groups of BIX01294 on HMT (G9a and GLP) inhibition. To this end, a scaffold-hopping strategy was conducted by replacing the quinazoline core with various bioisosteric moieties. The result is that two methoxyl groups, especially the 7-methoxyl group, are essential, since the replacement of two methoxyl groups with a dioxalone ring caused markedly impaired potency. Among the varied heterocycles, quinoline-derived derivative **25** gave the most potent affinity, which is largely owing to the basicity of protonated 1-N. Meanwhile, the 3-N atom of the quinazoline skeleton is not a necessity for G9a inhibition [41].

Noticing that there was still room for activity improvement for compound **24**, since *N*,*N*-dimethylbutane moiety did not entirely occupy the lysine-binding region, Jin et al. systematically examined how the length and disposition of 7-substituents of quinazoline derivatives affected the potency. As a consequence, compound **26** afforded the most potent G9a inhibition to date, with a 250-fold improved potency over the prototype BIX01294, as evinced in Figure 8 [42]. Albeit with excellent enzymatic activity, compound **26** presented lower cellular activity than BIX01294, which is largely attributable to its poor cell membrane permeability and low lipophicity (log *P* = 1.9). To overcome this bottleneck, this group further modified **26** by introducing various lipophilic groups, which has led to two promising hits. Compound **27** can effectively reduce the level of dimethylated H3K9 in many cell lines and shows high selectivity for G9a and GLP with low cytotoxicity [43]. Compound **28** not only exhibited high potency and selectivity for G9a and GLP, but has measurable DNMT1 inhibitory activity against DNA methyltransferase, implying that this compound has the potential to become a HMT/DNMT1 dual inhibitor [44]. However, **28** furnished a poor pharmacokinetic profile in an animal assay. The most probable reason might be due to the 2-cyclohexyl group, which is readily oxidized with the action of cytochrome p450 enzyme in vivo, leading to metabolic instability. Given that substituents at the 2-position of the quinazoline ring were well tolerated, this lab further optimized 2-substituents with the expectation of acquiring improved metabolic stability while maintaining high potency and high cellular activities. When 2-cyclohexyl was replaced by a 4,4-difluoropiperidinyl unit, the resulting compound **29** manifested the optimal performance and has been the first chemical probe against G9a and GLP used in animal studies [45].

Sbardella et al. employed a ring expansion strategy by replacing the quinazoline skeleton of **24** with a benzodiazepine framework to afford 1,4-benzodiazepine derivative **30**, which gave about a 35-fold improved DNMT1 potency, comparable G9a inhibition, lowered cytotoxicity and better metabolic stability than the hit **24**, indicating a broad prospect to be developed as a HMT/ DNMT dual inhibitor [46]. Similarly, Oyarzabal et al. utilized compound **28** as a hit to design more potent HMT/DNMT dual inhibitors. The quinazoline scaffold was initially refined, since the 3-N atom actually does not form any interactions with a protein backbone. As expected, both G9a and DNMT1 inhibitions of the obtained compound UNC-0638 (**31**) have been greatly improved. To guide further modifications, the binding mode of 31 was investigated by docking it into mouse DNMT1, as shown in Figure 9. The result evinced that the quinoline core properly occupies the NNMT1 backbone and 1-N forms a H-bonding interaction with Glu1269 (Glu1266 in human DNMT1), 7-O atom 4-NH and contributes to H-bonding contacts with Arg1315 (Arg1312 in human DNMT1) and Ser1233 (Ser1230 in human DNMT1), respectively. As to the 2-cyclohexyl unit, it did not form any interactions with protein, while the 4-terminal isopropyl piperidine group forms a hydrophobic interaction with Met1235 (Met1232 in human DNMT1) and the 6-methoxyl group points towards the 3′-direction of the DNA strand, implying that subsequent structural variations might be conducted at 2- and 4-positions. Guided by this docking result together with previously obtained SAR results, several more potent inhibitors **32**~**36** have been identified. However, comparing with their excellent G9a inhibition, there is still room for improving the DNMT1 affinity. Therefore, more detailed and comprehensive SAR information still needs to be replenished.

Chaetocin (**37**), a fungal mycotoxin from Chaetomium minutum and a specific Suv39H1 inhibitor that works as a competitive inhibitor for SAM, as exhibited in Figure 10, was capable of inducing 86% latent proviruses from HIV-1 infected HAART-treated patients without affecting T-cell function [47]. By contrast, the control HMTI BIX01294 gave slightly decreased (80%) potency under the same conditions [48]. Furthermore, EZH2 inhibitors, such as GSK343 (**38**), 3-deazaneplanocin A (DZNep, **39**) and tazemetostat (EPZ-6438, **40**), and EHMT2 inhibitor UNC-0638 were also reported to display strong anti-latency-reversing activities. Intriguingly, two selective EZH2 inhibitors GSK343 and tazemetostat furnished more effective latency-reversing activities than the broad EZH2 inhibitor DZNep, further proving the direct association between specific EZH2 inhibition and HIV-1 reactivation. In addition, the combination of any of these HMTIs, as displayed in Figure 10, with HDACI (e.g., SAHA, vorinostat) procured a strong synergistic HIV-**1** reactivation effect [49].

Actually, with advancement in various molecular biology techniques, more and more previously unrecognized histone modification-associated factors underlying HIV-1 latency are being disclosed in addition to the widely investigated EZH2, G9a and GLP. For example, since the latency-related histone mark H3K27me3 is catalyzed by polycomb repressive complex 2 (PRC2), while embryonic ectoderm development (EED) is a principle component of PRC2, inhibition of EED is therefore inferred to be largely beneficial to latent provirus reactivation similarly to other HMTIs. James et al. proved that two EED inhibitors (EEDIs), A-395 (**41**) and EED226 (**42**), not only presented promising HIV-1 reactivation efficacies after treating alone but also gave an additive effect in latently infected 2D10 cells when in combination with EZH2 inhibitors, GSK343 or UNC1999 (**43**), further confirming that PRC2-mediated components, such as EZH2, EED and SUZ12, can serve as attractive targets to develop more types of LRAs [50].

Besides, Ott and co-workers disclosed a previously undiscovered lysine methyltransferase, SET and MYND domain-containing protein 2 (SMYD2), which is also a potent target for identifying latency-reversal agents. SMYD2, which works as an epigenetic co-repressor, is found to be tightly linked with HIV-1 latency by inducing monomethylation on histone H4K20me1 at the HIV-1 5′-LTR region, leading to repression of proviral transcription. Accordingly, H4K20me1 can also act as an important histone mark in addition to H3K27me3 and H3K9me2/3. The fact that effective reactivation of latent proviruses with a SMYD2 inhibitor AZ391 (**44**) in CD4^+^ cells further supported the inhibition of the SMYD2’s catalytic activity being directly correlated with HIV-1 reversal activity [51]. Meanwhile, given that H4K20me1 is recognized by a chromatin “reader” protein lethal 3 malignant brain tumor 1 (L3MBTL1), leading to chromatin compaction and consequent transcriptional silencing, L3MBTL1 thereby becomes a possible HMT pathway-related target for exploiting LRAs, although its exact role in modulating HIV-1 latency still needs further verification [52].

Taken together, among the proposed HMT-associated latency factors, EZH2 has stronger associations with HIV-1 reactivating efficacy compared with other HMT-related factors, including but not limited to G9a, SUV39H1, EED, SMYD2 and L3MBTL1. Hence, specific and effective EZH2 inhibitors are strongly encouraged.

### 2.4. Protein Kinase C Activators

The protein kinase C (PKC) family of serine/threonine kinases plays an essential role in reactivating latent HIV-1 through activation of the NF-κB signaling pathway [53]. A host of small molecules derived from natural sources, including but not limited to phorbol esters, ingenol esters, ingols, jatrophanes, and various macrolides, have been proposed to reactivate HIV-1 in latently infected CD4^+^ T cells, as displayed in Figure 11. Among them, ingenol derivatives phorbol 12-myristate 13-acetate (PMA, **45**) and 12-deoxyphorbol-13-acetate (prostratin, **46**) are two most well-known activators of PKC with nanomolar PKC activation affinities. Although PMA reactivates latent HIV-1 by activating T cells, its clinical utility was limited by tumor-promoting risk and other serious side effects, such as mitotic dysfunction and chromosomal aberrations [54].

Unlike PMA, prostratin, which is isolated from the poisonous New Zealand plant Pimela prostrata, is not a tumor promoter and does not induce cell proliferation by itself. Besides, prostratin can inhibit PMA-induced tumor promotion in a mouse model, illustrating a broad scope in clinical application. Prostratin not only reactivates latent HIV-1 in vitro in a PKC-dependent NF-κB activation manner, but also down-regulates the expressions of HIV-1 receptor CD4 and co-receptor CXCR4, thus avoiding the novo infection of CD4+ cells. However, since prostratin causes overall T cell activation, just as PMA does, further investigation is still needed in order to interpret the suitability of this compound for use in humans [55].

Another two ingenol derivatives, phorbol 13-stearate (P-13S, **47**) and 12-Deoxyphorbol 13phenylacetate (DPP, **48**), also presented attractive HIV-1 reactivation potencies. P-13S effectively activates HIV-1 gene expression in the Jurkat-LAT-GFP latency model, with at least a 10-fold increase in potency over that of prostratin. Interestingly, P-13S activates PKC by inducing a translocation of PKC isotypes α and δ to cellular compartments, which is distinctly different from that of prostratin and PMA [56]. As to DPP, a non-tumour-promoting phorbol ester isolated from the West African “candle plant” Euphorbia poissonii and the Moroccan succulent E. resinifera Berg, was reported to induce HIV-1 gene expression in latently infected ACH-2 cells at a 20–40-fold lower concentration than prostratin [57].

The ingenol analogue ingenol-3-angelate (PEP005, **49**) that was isolated from Euphorbia peplus was validated to reactivate latent HIV-1 through a PKC-dependent NF-κB pathway. Moreover, this compound was able to prevent new rounds of viral infection after HIV-1 reactivation through down-regulating the expression of the HIV-1 receptors CD4 and CXCR4 [58]. The ingenol derivative EK-16A (**50**) that was isolated from Euphorbia kansui displayed 200-fold more potent efficacy than prostratin in reactivating latent proviruses in vitro and ex vivo with minimal cytotoxicity on cell viability. Besides, EK-16A could induce synergistic effects with multiple types of LRAs, such as DNMTI 5-Aza, BETIs JQ1 and I-Bet151, as well as HDACIs vorinostat and romidepsin in latently infected cell lines J-Lat 10.6 and 6.3, without any influence on T-cell activation. Mechanistically, EK-16A works as a PKCγ activator to promote HIV-1 transcription HIV-1 reactivation via the activation NF-κB pathway and also to facilitate HIV-1 elongation via the stimulation P-TEFb pathway [59]. Later, three ingenol derivatives, viz. EK-1A (**51**), EK-5A (**52**) and EK-15A (**53**), were isolated from Euphorbia kansui, and not only demonstrated latent reactivation efficacies in vitro and ex vivo at nanomolar concentrations but also could inhibit acute HIV-1 infection via down-regulation of the expression of CCR5 and CXCR4, two cell surface HIV co-receptors. Additionally, these three ingenol derivatives have a synergy with 5-Aza, SAHA, JQ1 or prostatin with little cellular toxicity in T-cells [60]. These data suggested ingenol derivatives derived from Euphorbia species might have great prospects for being developed into successful chemotherapeutic LRAs.

Macrocyclic jatrophane diterpenoid compound SJ23B (**54**), which is isolated from a Mediterranean plant specimen, E. hyberna, is an activator of PKCα/δ. It could reactivate latent HIV-1 via activation of the NF-κB pathway at a nanomolar level (EC_50_ = 50 nM), which is at least 10 times more potent than prostratin. Moreover, SJ23B is not a tumor promoter and displayed strong in vitro anti-HIV-1 activity [61].

The epimeric *N*,*N*-dimethylvalinoyl-4α-4-deoxyphorbol derivatives **55** and **56** were isolated for the first time from a medicinal Mexican Croton, and were later found to be potent and isoform-specific activators of PKC with promising HIV-1 reactivating activity [62]. 3,12-Di-O-acetyl-8-O-tigloyl-ingol (**57**) that was isolated from Euphorbia lactea, a plant that produces latex with anti-inflammatory activity, was reported to antagonize HIV-1 latency through a PKC-dependent pathway [63]. Likewise, 8-methoxyingol 7,12-diacetate 3-phenylace (**58**) which was derived from the latex of Euphorbia officinarum, could also reactivate HIV-1 latency with an EC_50_ < 25 µM [64].

Macrolides bryostatins that are isolated from bryozoan are a family of effective PKC activators, among which bryostatin-1 (**59**) exhibited the most powerful PKC activating activity at a nanomolar concentration and has been the only PKC agonist that entered clinical evaluation as an effective LRA candidate. Bryostatin-1 not only exhibited significant HIV-1 reactivation efficacy in human astrocytes via a NF-κB and PKC-dependent mechanism, but induced a remarkable decrease in viral production and amyloid beta (Aβ) deposition in myeloid cells. Bryostatin-1 can also activate the mitogen-activated protein kinase (MAPK) pathway and down-regulate the expressions of HIV-1 co-receptors CD4 and CXCR4 without triggering global T cell proliferation, and it has synergy with HDACIs in reactivating latent HIV-1 [65]. All these beneficial properties together with the fact that bryostatin-1 was capable of avoiding de novo infection in HIV-1 in susceptible cells [66] makes it a promising adjunct for the treatment of HIV-1 brain infection [67].

Enlightened by the promising therapeutic potential and SAR outcomes of bryostatin-1, Stone and co-workers prepared two bryostatin-1 analogs, compounds **60** and **61**, which offered comparable to higher PKC-binding affinities compared to the prototypes bryostatin-1 and PMA [68]. In addition, the semi-synthetic ingenol ester **62** was reported to exert HIV-1 reactivation efficacy by activating PKCs and up-regulating the positive transcription elongation factor b (P-TEFb), thus promoting both transcription initiation and elongation of viral genes. Given that ingenol analogues have rich natural resources, compound **62** is therefore a promising LRA candidate to be utilized in clinical practice [69]. Gnidimacrin (**63**), a diterpenes PKC activator, can potentially activate latent HIV-1 viruses, with about 10 times more potency than HDACI SAHA at an effective concentration as low as the picomolar level. It is especially noteworthy that **63** can significantly reduce the size of a latent reservoir by decreasing the amount of latently infected cells at a concentration that does not cause overall T cell activation or stimulate production of inflammatory cytokines, indicating its promising clinical application [70].

BL-V8-310 (**64**), a benzolactam-related PKC activator, was proved to effectively reactivate latent HIV-1 in latently infected ACH-2 and J-Lat cell lines. Moreover, combining BL-V8-310 with BRD4 inhibitor JQ1 not only showed synergistic latency-reversing activity but also decreased the influence on cytokine secretion from CD4^+^ T cells induced by BL-V8-310 alone [71].

In brief, PKC activators are still attractive LRAs which are particularly effective in combination with other LRAs with different mechanisms.

### 2.5. BET Inhibitors (BETIs)

Among the four members (BRD2, BRD3, BRD4, and BRDT) of the bromodomain and extra-terminal (BET) protein family, BRD4 is the most concerned and widely studied subtype. The primary structure of BRD4 comprises two highly conserved N-terminal tandem bromodomains (BD1 and BD2) that are responsible for recognizing the acetylation status of lysine residues on histone tails and other proteins, an ET domain, and a C-terminal domain (CTD). Bromodomain protein contains four antiparallel α-helices (αZ, αA, αB, αC) and two hydrophobic loops (ZA loop and BC loop). ZA, BC, and αZ form an essential *N*-acetylated lysine-binding hydrophobic pocket (Kac site). Besides, a hydrophobic region ‘WPF shelf’ bounded by Trp/Pro/Phe residues (W81, P82, F83) is also important for enhancing BRD4 binding affinity [72].

Acting as an interaction partner with P-TEFb, BRD4 can competitively bind P-TEFb with Tat protein, an exclusive transcriptional transactivator of HIV-1 viruses, thus leading to the silencing of HIV-1 gene transcription. Hence, inhibition over-expression of BET/BRD4 will promote the recruitment of P-TEFb by Tat and subsequent viral transcription elongation, resulting in dissociation of the BET and P-TEFb complex and consequent provirus reactivation [73]. Chemical structures of several representative BET inhibitors (BETIs), as displayed in Figure 12, are being currently tested in various latently HIV-1-infected evaluation models.

Triazolothienodiazepine derivative (+)-JQ1 (**65**) is the first BETI discovered by a high-throughput screening protocol, which possesses many good drug-like properties, such as excellent cellular potency, high selectivity for the BRD4 isoform of the BET family, synthetic accessibility, low off-target possibility and a good pharmacokinetic profile. However, JQ1 was less efficient in reactivating latent HIV-1 viruses when used alone and exerted severe cytotoxicity during prolonged treatment [74]. Since the benzodiazepine fragment of JQ1 is validated to be a key pharmacophore that contributes to a good BRD4 affinity, a spectrum of benzodiazepine-derived BETIs was successively identified, including but not limited to OTX015 (**66**), CPI-203 (**67**), MS417 (GTPL7512, **68**) and I-BET762 (GSK525762, **69**).

OTX015 (**66**) could effectively reactivate latent HIV-1 in various latency models with 1.95~4.34 times improved efficacy over that of JQ1. More interestingly, OTX015 can also increase CDK9 occupancy at HIV-1 promoter, which in turn phosphorylates RNA polymerase II (Pol II) to enable productive elongation and generate full-length HIV transcripts [75].

CPI-203 (**67**) is a novel cell-permeable BRD4 inhibitor with good bioavailability when administered orally. Notably, in contrast to JQ1, CPI-203 showed more potent HIV-1reactivation efficacies in various latently infected cell lines and significantly decreased cytotoxicity. CPI-203 also exerted a synergistic effect and alleviated prostratin-induced ‘cytokine storm’, a severe and systemic production of inflammatory cytokines in response to endotoxemic shock, when combined with PKC activators (e.g., prostratin, ingenol-B and bryostatin-1) in the reactivation of latent HIV-1 [76]. MS417 (**68**) and I-BET762 (**69**) could reactivate HIV-1 from latency in vitro via a P-TEFb-dependent but Tat-independent mechanism, implying BRD2 together with BRD4 cooperatively participate in HIV latency [77].

BETIs PFI-1 (**70**) and RVX-208 (**71**), whether used alone or in combination with other types of LRAs, exhibited strong HIV-1 reactivation activities through up-regulation of the phosphorylation of CDK9 Thr-186 to increase the expression of P-TEFb in latently infected Jurkat T cells, thus effective activation of HIV-1 transcription. Besides, these two BETIs could also activate HIV-1 transcription in resting CD4^+^ T cells in cART-treated patients without causing aberrant activation of global immune cells, strengthening the therapeutic values of BETIs in anti-HIV therapy by effective combating HIV-1 latency [78].

Mivebresib (**72**) has already been advanced to phase I clinical trials for treating relapsed/refractory acute myeloid leukemia [79]. I-BET151 (**73**) could preferentially reactivate HIV-1 gene expression and efficiently increase HIV-1 transcription in monocytic cells, but not in typically HIV-1-infected CD4^+^ T cells, in cART-treated humanized mice, via a CDK2-dependent mechanism [80].

8-Methoxy-6-methylquinolin-4-ol (MMQO, **74**), was identified from a virtual screening that was able to reactivate latent proviruses in vitro and ex vivo by functioning as a BRD4 inhibitor. In addition, MMQO also could potentiate the effects of PKC activators or HDACIs on HIV-1 reactivation. Owing to its minimalistic structure, MMQO might act as an ideal hit for further chemical modification [81]. UMB136 (**75**), an imidazo [1,2-*a*]pyrazine-derived BETI, exhibited elevated HIV-1 reversal activity compared to JQ1 in multiple latently infected cell models. Besides, UMB136 was able to synergize with PKC activators (e.g., bryostatin-1 and prostratin), and increase viral production and HIV-1 transcription by releasing P-TEFb [82].

The BRD4 selective inhibitor ZL0580 (**76**) was capable of reactivating latent HIV-1 and restraining viral replication in CNS reservoirs, indicating the therapeutic potential in treating neuroinflammation or related neurological disorders in HIV-infected individuals [83]. CPI-637 (**77**), a BRD4 and TIP60 dual inhibitor, could efficiently reactivate latent proviruses in vitro. Meanwhile, CPI-637 was also able to alleviate overall T cell activation and prevent viral spread to uninfected CD4^+^ T cells without an obvious toxic effect. Interestingly, CPI-637-mediated TIP60 inhibition, in turn, stimulated BRD4 dissociation from the HIV-1 5′-LTR promoter, causing more effective binding of Tat protein with P-TEFb in comparison with the BRD4 inhibition alone. These data indicated that development of dual-target LRAs, including but not limited to BRD4/TIP60 inhibitors, might be a more promising remedy to effectively eliminate latent reservoirs [84].

According to the X-ray co-crystallographic structures of BETIs-bound BRD4, the vast majority of the aforementioned BETIs share two key pharmacophoric features: a ‘head moiety’ that bears an H-bond donor and/or acceptor to form hydrogen bonds with two key amino residues (Asn140 and Tyr 97), by mimicking the function of the acetyl O atom of ABBV-075, and appropriate hydrophobic substituents to spatially accommodate the ‘WPF shelf’ and ‘ZA channel’, aiming to enhance potency and selectivity towards BRD4 by forming intensive hydrophobic contacts [85].

To identify more potent BETIs, Burns and co-workers believed an appropriate Kac-mimicking pharmacophoric framework, which is capable of providing interactions with the pivotal Kac domain and meanwhile offers strong H-bonds with the key residues in this region, plays a crucial role in achieving desirable BET-binding affinities. To this end, they employed a scaffold-hopping strategy by replacing 6*H*-thieno [3,2-*f*][1,2,4]triazolo [4,3-*a*][1,4]diazepine core of JQ1 with a 1,2,3-triazolobenzodiazepine framework, which was supposed to be capable of forming more favorable binding interactions with target protein. It indicated that, as anticipated, the resulting 1,2,3-triazolobenzodiazepine skeleton (**78**) gave a nearly 3-fold increase in BRD4-binding affinity. The introduction of a 2-aminopyridine fragment on the central benzene ring (**79**) resulted in further activity improvement. To figure out the possible reason for activity improvements of the 1,2,3-triazolobenzodiazepine nucleus, binding patterns of JQ1 and **79** complexed with BRD4 (PDB entry: 5 UVV) were compared by a molecular modeling method. As evinced in Figure 13, although both 76 and reference JQ1 adopt similar spatial orientation by docking well into three key binding domains (Kac, ZA, WPF) of BRD4, **79** formed two H-bonds between the 2-aminopyridine portion and two residues (Met398 and Met425), with distances of 2.3 Å and 2.8 Å, respectively. By contrast, JQ1 formed a relatively weak H bond with Tyr390, with a distance of 3.1 Å. Collectively, compound **79** with the new skeleton forms stronger H-bond contacts, which is supposed to largely account for its elevated activity [86].

I-BET151 (**80**) is another representative BETI, in which 1*H*-imidazo [4,5-*c*]21uinoline-2(3*H*)-one core is a key factor in affecting HIV-1 reactivation potency by functioning as a mimic of acetylated lysine. The co-crystal structure of I-BET151 complexed with BRD4 (BD1), as evinced in Figure 14, revealed that the 3,5-dimethylisoxazole part forms a direct H-bond with Asn140 (3.2 Å) and water-bridged hydrogen-bonding interactions with Tyr97 (2.0 Å) in the acetylated lysine binding pocket (Kac site). The 1H-imidazo [4,5-*c*]21uinoline-2(3*H*)-one nucleus stretches into the hydrophobic ZA channel of protein, and the N atom on the central pyridine ring forms a water-mediated hydrogen bond interaction with Phe83 (2.7 Å). The pyridine ring interacts with the surface of the hydrophobic WPF shelf, which is formed by W81, P82, M149 and I146 residues of BRD4.

Based on this binding information, Wang’s lab employed a ‘scaffold-hopping’ strategy to substitute a [6,6,5]tricyclic 5*H*-pyrido [4,3-*b*]indole (γ-carboline) motif which has a balanced physiochemical profile for the 1*H*-imidazo [4,5-*c*]21uinoline-2(3*H*)-one core of I-BET151, with the expectation of achieving similar BRD4-binding affinity. Among the newly obtained derivatives, compound **81** containing a 3-cyclopropyl-5-methyl-1*H*-pyrazole fragment gave the best potency, particular an improved BRD4(2)-binding affinity compared with the reference JQ1, while the replacement of this moiety with a smaller Cl atom (**82**) impaired potency [87].

To gain more insight into the activity differences, binding modes were conducted and compared between I-BET151 and compound **81** by using the molecular docking method. As evinced in Figure 15, although both **81** and I-BET151 adopt very similar spatial orientation and both bind with two key residues (Gln85 and Asn140), **81** offered stronger H-bonding interactions with these two amino acids than I-BET151, which might explain their binding differences to a great extent.

In summary, although the exact molecular mechanism of BETIs in reactivating latent proviruses is still not fully understood, the successful development of various BETIs highlights the importance of BET/acetyl-lysine inhibition in regulating HIV-1 transcription and latent reactivation. Nevertheless, so far, none of the BETIs have entered clinical phases as LRAs, primarily attributable to the in vivo inefficiency with no effect on eliminating both latent reservoirs and HIV-1 viremia, although the in vitro latency-reversing activities were satisfactory. This fact leaves the clinical implementations of BETIs as LRAs with a long way to go still.

### 2.6. P-TEFb Activators

P-TEFb, which is composed of cyclin T1 (CycT1) and cyclin-dependent kinase 9 (CDK9), is a CDK9-cyclin T1 heterodimer and is an important part of the super elongation complex (SEC) used by the viral-encoded Tat protein to activate HIV transcription. In this sense, P-TEFb is an indispensable cellular cofactor for Tat protein and an essential elongation factor to realize efficient viral transcriptional elongation and expression. In this regard, P-TEFb activators were able to work as LRAs to reactivate latent viruses [88].

Ye et al. isolated at least five kinds of short-chain fatty acids (SCFAs), namely, butyric acid, isobutyric acid, isovaleric acid, propionic acid and acetic acid, from the saliva and plasma of HIV-infected patients with severe periodontitis undergoing HAART treatment. The subsequent study revealed that, except for acetic acid, the other four SCFAs have the capability to potently reactivate latent HIV-1 in both latent Jurkat cells and resting CD4^+^ T-cells by activating P-TEFb as well as inducing histone modifications in a dose-dependent manner. Among these five SCFAs, butyric acid presented the strongest activity, while isobutyric acid gave the weakest effect, and the combination of any two SCFAs exhibited an additive effect [89].

PR-957 (**83**), also named as ONX-0914, as displayed in Figure 16, might be a promising LRA candidate with a host of favorable properties. It was found to strongly activate latent viruses in vitro and in vivo by activating P-TEFb without causing overall abnormal immune activation of T cells. It also exhibited relatively low cytotoxicity at a tolerable level and could reduce the expression of HIV receptor CD4 and co-receptors CXCR4 and CCR5. Moreover, it manifested a synergistic effect when combined with known LRA, PKC activator prostratin, and could reduce prostratin-induced T-cell activation. Most notably, it displayed no interference with other anti-viral drugs, thus guaranteeing the efficient implementation of the “shock and kill” strategy and effective elimination of newly produced viruses [90].

Hexamethylene bisacetamide (HMBA, **84**), which has been dismissed as anticancer candidate in clinical anti-leukemic evaluation, has recently revived interest in anti-HIV therapy due to its remarkable capability in reactivating latent HIV-1 proviruses, whether used alone or in combination with PKC activator prostratin in chronically infected CD4^+^ T cells [91]. The reactivation occurs at a transcription level and is independent of NF-κB but transiently activates Akt via the inhibition of phosphatidylinositol-3-kinase (PI3K) and relies on Sp1-binding sites in the HIV promoter to recruit P-TEFb to promote the reactivation of viral production from latency [92]. Chalcone analogue Amt-87 (**85**) has been proved to significantly reactivate latent HIV-1 provirses and synergistically work with known LRAs, such as PKC activator prostratin and BETI JQ1 via activation of P-TEFb, indicating its development perspective as a potent LRA [93].

### 2.7. Polo-like Kinase 1 (PLK1) Inhibitors

Acting as a highly conserved Ser/Thr kinase in eukaryotes, PLK1 is proved to be a promising target in identifying potent LRAs. PLK1 inhibitors are expected to have doubled function by not only effectively reactivating latent proviruses but also promoting the death of reservoir cells, based on the fact that PLK1 plays an essential role in HIV-1 Nef-mediated survival of CD4^+^ T cells [94]. Besides, PLK1 also has a tight correlation with BRD4. On one hand, both BRD4 and PLK1 are able to interact directly with P-TEFb to regulate HIV-1 LTR transcription; on the other hand, BRD4 can also be phosphorylated by PLK1 [95], which perhaps explains why many PLK1 inhibitors, such as compounds 5-methyl-7,8-dihydropteridin-6(5*H*)-one derivatives BI-2536 (**86**) and BI-6727 (volasertib, **87**), as diagrammed in Figure 17, also function as BRD4 inhibitors.

Acting as dual PLK1/BET inhibitors, pteridine-derived compounds **86** and **87** could significantly reactivate silenced HIV-1 proviruses in two latently infected cell lines, ACH2 and U1, at both the mRNA and protein level. Although these two compounds demonstrated more potent PLK inhibitory activities, the latent reactivating effects were directly associated with BET inhibition, further confirming the important role of BETIs in developing promising LRAs. Additionally, BI-2536 could synergistically reactivate HIV-1 proviruses when combined with HDACI SAHA or PKC activator prostratin, in PBMCs from HIV-1-infected patients, verifying the prospective therapeutic value in anti-viral treatment [96].

### 2.8. CCR5 Antagonist

Maraviroc (MVC, **88**), as shown in Figure 18, a selective CCR5 (C-C Motif Chemokine Receptor 5) antagonist with broad-spectrum antiretroviral efficacy that has been used in treatment of HIV-1 infection, was recently validated to be an attractive LRA by inducing NF-κB activation as a result of specific binding of CCR5, whether used alone or in combination with PKC activator bryostatin-1 [97]. However, despite the favorable safety profile in humans [98] as well as positive latent reactivation outcomes in phase II trial, maraviroc could not efficiently reduce the size of latent reservoirs, given the fact that viruses rebounded quickly after antiretroviral therapy was discontinued [99]. All these evidence strongly support that CCR5 antagonists might be successfully developed into effective anti-HIV drugs, which not only function as antiretroviral agents to suppress HIV-1 replication but also act as LRAs to reactivate latent proviruses [100].

### 2.9. Inhibitor of Apoptosis Protein (IAP) Antagonist

Xevinapant (Debio 1143, AT-406, **89**), as displayed in Figure 19, an effective chemo-radio-sensitizer as a first-in-class oral IAP antagonist, currently being assessed in phase II clinical trial for the treatment of squamous cell carcinoma of the head and neck (SCCHN), was recently proved to reverse HIV-1 latency in various latently infected T cell lines via the induction of a non-canonical NF-κB pathway, resulting in the enhancement of HIV-1 transcription [101]. Besides, an obvious synergistic latency-reversal activity was achieved when combined use of IAP antagonist AZD5582 (**90**) and BETI I-BET151, rather than largely due to these two LRAs both acting on host cell transcriptome [102]. The favourable latency-reversal efficacies together with the manageable safety profiles may enable IAP antagonists to be developed as promising LRAs.

### 2.10. Phosphatidylethanolamine-Binding Protein 1 (PEBP1) Inhibitor

Zhu et al. found that the PEBP1 gene, also named as RKIP (Raf kinase inhibitor protein) is tightly correlated with the establishment of a latent reservoir as well as the suppression of viral replication by employing a CRISPR-based genetic screen technique. The subsequent depletion of the PEBP1 gene significantly reactivates latent proviruses by regulating the NF-κB pathway, while the up-regulation of PEBP1 expression with epigallocatechin-3-gallate (EGCG, **18**) inhibits latency reversal by preventing nuclear translocation of NF-κB. These results provided directly proofs that PEBP1 inhibitors might be a type of promising LRAs in effective controlling HIV-1 reservoirs [103].

### 2.11. Proteasome Inhibitor (PI)

Proteasome inhibitor (PI) thiostrepton (TSR, 91), as demonstrated in Figure 20, a naturally derived thiazole-containing oligopeptide antibiotic, was recently found to effectively reactivate latent HIV-1 proviruses in vitro and ex vivo with low cytotoxicity. TSR also displayed a significant synergistic effect with prostratin, bryostatin-1 or JQ1 without affecting overall T cell activation or inducing dysfunction of CD8^+^ T cell, indicating the prospective application as an effective LRA candidate. Mechanistic study revealed TSR activated P-TEFb and NF-κB pathways by up-regulating heat shock proteins (HSPs), resulting in viral reactivation [104].

By using the CRISPR interference-based screening technique, Zhou et al. proved two PIs, bortezomib (PS-341, **92**) and carfilzomib (**93**), could effectively reactivate latent HIV-1 by strongly synergizing with different types of LRAs without inducing CD4^+^ T cell activation or proliferation. Mechanistic investigation showed that a proteasome inhibitor could promote Tat-transactivation by up-regulating the expressions of ELL2 and ELL2-containing super elongation complexes (ELL2-SECs), leading to reactivation of latent proviruses [105]. Mechanistic results from Liu and co-workers demonstrated that PIs not only could reactivate latent HIV-1 by HSF1-mediated recruitment of complex of HSP90 and P-TEFb, but also may facilitate the human body to the virus by enhancing host innate immune responses [106]. Evidence from Dougherty et al. suggested that PIs, exemplified by bortezomib, clasto-lactacystin β-lactone (CLBL, **94**) and MG-132 (**95**), work as bifunctional antagonists of HIV-1, exerting anti-latency by reactivating latent HIV-1 and anti-replication by inhibiting HIV-1 infectivity [107]. All these data revealed that proteasome can function as a promising and targetable target in developing effective anti-HIV-1 agents [108].

In addition to proteasome, HSF1 and HSP90, which participate in HIV transcription elongation, can also serve as attractive therapeutic targets in developing potent LRAs. Particularly, HSF1 that is tightly associated with HIV-1 5′-LTR might be a targetable target for identifying promising LRAs based on the fact that the majority of LRAs with various mechanism of actions might reactivate latent HIV-1 via an HSF1-involved pathway rather than the previously known pathways [109].

### 2.12. Toll-like Receptor (TLR) Agonist

Toll-like receptors (TLRs) are transmembrane pathogen-recognition receptors that function as the sentinels of host defense by recognizing a spectrum of pathogen-related molecular components in bacteria, viruses, protozoa and fungi as well as molecules released during cell damage or death. The TLR family contains more than 13 members, of which only 10 (TLR1-TLR10) have been identified in humans [110].

Since TLR members are mainly existing and functional in CD4^+^ T cells, where the majority of the latent reservoir is harboured, TLR agonists are thereby believed to hold great promise to successfully reactivate latent proviruses by stimulating downstream pathways [111], such as AP-1, NFAT, IRFs as well as NF-κB, a well-acknowledged regulator of HIV transcription that has a tight correlation with HIV-1 latency [112,113].

Moreover, TLRs also have high hopes of eliminating activated latent cells by exerting immunologic cytotoxicity, promoting antiviral responses and non-specifically activating T lymphocytes. All these properties render immunostimulatory TLR agonists unique and promising LRAs among the identified LRAs to date, also known as next-generation LRAs [114].

TLR signalling, in particular TL-7/8/9 and TL-1/2 activation, has shown to be notable LRA with multifactorial characteristics. For instance, resiquimod (R848, **96**), as exhibited in Figure 21, a imidazoquinoline-based TLR7/8 agonist, showed latency-reversing activity in latently infected cell lines U1 and OM10 by inducing p24 expression [115]. Moreover, resiquimod was capable of diminishing the size of latent reservoirs in HIV-1 infected patients and meanwhile preventing virus replication, suggesting potential clinical value [116]. Vesatolimod (GS-9620, **97**), a dihydropteridinone-derived TLR7 agonist (EC_50_ = 291 nM) could effectively reactivate latent proviruses in PBMCs from HIV-1-infected individuals and also improve immune effector functions at a safe and tolerable clinical dose [117]. Rochat et al. supposed the combination of transcriptional enhancers with a TLR agonist would be more efficient in reversing latency at various levels by generating a Th1 supportive milieu and meanwhile triggering the innate immune system. It was revealed that the combination of thiazoloquinolone-based TLR8 agonist CL075 (3M002, **98**) with PKC agonist prostratin resulted in significantly augmented latency-reversing activity compared with any single compound, whether in primary cells of HIV-1-infected individuals or on a coculture of J-lat and MDDCs (monocyte-derived dendritic cells). These data implied that the combined use of two LRAs, by which one triggers directly the transcriptional pathway and the other stimulates indirectly the immune system, would be a practicable remedy for efficiently reversing HIV-1 latency [118]. SMU-Z1 (**99**), an imidazole-based TLR-1/2 agonist, exhibited excellent latency-reversing potency in HIV-1-infected cells via activation of NF-κB and MAPK pathways in vitro and in PBMCs from HIV-1-infected individuals ex vivo. Besides, SMU-Z1 was also capable of promoting degranulation and gamma interferon (IFN-γ) production in NK cells, as well as increasing TNF-α production in PBMCs without triggering overall T cell activation, indicating the prospect application in eradicating HIV-1 [119].

Considering that some TLR signaling, particularly TLR3, are distributed in microglia cells, which constitute the major HIV-1 reservoir in the brain, Karn and co-workers screened a panel of TLR agonists on different HIV-1 infected cell lines. The results revealed that TLR3 agonists, as anticipated, could efficiently reactivate viral transcription in HIV-1-infected microglia cells, rather than monocytes or T cells, via a previously unreported IRF3, but not the most common NF-κB-mediated mechanism, providing evidence for the therapeutic potential of TLR3 agonists in the treatment of HIV-1 infection in the central nervous system [120].

Despite many beneficial profiles, the precise mechanisms by which the TLR agonists reactivate latent proviruses and modulate immune responses, however, remain incompletely understood. Therefore, more efforts should be made to figure out the detailed mechanisms and identify more robust immunostimulatory TLR agonists against HIV-1 [121].

### 2.13. Unclassified LRAs with Undefined Biotargets

#### 2.13.1. AV6 and Analogues

Quinoline-derivative antiviral 6 (AV6, **100**), as displayed in Figure 22, was identified from a cell-based high-throughput screening protocol, and could reactivate latent HIV-1 in different cell-based latency models without inducing global T-cell activation or proliferation, and could also act synergistically with a HDACI valproic acid (VA) to achieve an additive HIV-1 reactivation [122]. Considering the definite efficacies of HDACI (especially class I HDACI) and AV6, used either alone or in combination, in reactivating latent proviruses, Fang and co-workers expected to develop novel HDACI-derived LRA chemotypes to gain enhanced induction of HIV-1 gene expression. To this end, they investigated various linker lengths and ZBGs, while leaving the *N*-phenylquinoline skeleton of AV6 unchanged in order to act as a cap region. Among the obtained AV6 derivatives, compounds **101** and **102** exhibited the most potent in vitro latency-reversing activities in J-Lat A2 cells. A preliminary mechanistic study confirmed these two compounds function as dual HIV-1 reactivators by inhibiting HDAC2 isoform and meanwhile facilitating the release of P-TEFb, and 94-dependent HIV-1 gene expression could be obstructed by an immunosuppressant tacrolimus (FK506). However, the anticipated synergetic effect did not occur in combined use of **102** and AV6 [123].

#### 2.13.2. 2-Acylaminothiazole

2-Acylaminothiazole (**103**a), as shown in Figure 23, was able to reactivate HIV-1 gene expression with modest activity (EC_50_ = 23 μM) in a high-throughput screening protocol using a dual luciferase reporter cellular assay, which thereby became an ideal hit to develop more chemically diversified LRAs. Among a selection of 2-acylaminothiazole derivatives, **103**b–c, **103**a in particular, gave comparable to increased activity compared to controls, vorinostat and JQ1, in latent CD4^+^ T cells isolated from cART-treated patients, implying that the introduction of electron-withdrawing groups (e.g., cyano, trifluoromethyl) is favorable to the elevation of HIV-1 reactivation efficacy. Furthermore, synergy could be observed between **103**c and BETI JQ1 in various latent HIV-1 cellular models, indicating the prospects as effective LRAs for treating HIV-positive individuals. However, the exact cellular target(s) of these derivatives were still unclear [124].

#### 2.13.3. Dihydropyranoindole Derivative

Pyranoindole derivative GIBH-LRA002 (**104**), as exhibited in Figure 24, was discovered from a high-throughput screening protocol in the J-Lat cell model, which has a decent HIV-1/SIV reactivation efficacy in resting CD4^+^ T cells from both chronic SIV-infected rhesus macaques and HIV-1 infected patients. Meanwhile, it possesses relatively low cytotoxicity without causing overall T cell activation, thus avoiding new viral infections, indicating that GIBH-LRA002 might be an attractive candidate for anti-HIV treatment [125].

#### 2.13.4. Carbazole Derivative

Curaxins CBL0137 (**105**), as manifested in Figure 25, has been described as a potent FACT (facilitates chromatin transcription) inhibitor, which exerts anticancer efficacy by suppressing the NF-κB pathway and activating tumor suppressor gene p53 [126]. As previous work has proved the NF-κB pathway plays an important role in disrupting latent HIV-1, while the depletion of FACT also could reactivate HIV-1 proviruses, CBL0137 was thereby submitted to the anti-latency evaluation in post-integrated latency cells, JLAT6.3 and CA5. The result evinced that CBL0137 alone could not only sufficiently potentiate TNF-α stimulated HIV-1 transcription without inducing any cytotoxicity, but also reactivate latent reservoirs in PBMCs from HIV-1-infected patients, suggesting the potential clinical application as an appealing LRA candidate [127].

#### 2.13.5. Benzazole Derivative

Benzazole derivative **106**a (see Figure 26) was found to have well-defined HIV-1 reactivation activity and a favorable pharmacokinetic profile in a high-screening protocol in 24STNLSG cells, which thereby was a good hit for further chemical optimization. Among the synthesized benzazole derivatives, **106**b gave the best HIV-1 reactivation effectiveness and the lowest cytotoxicity, which could induce proviral transcription in several latently infected cell types without affecting overall T cell activation. However, the 4-NH2 containing benzazole derivative **106**c provided decreased HIV-1 latency potency in a latently infected ACH-2 cell line. Preliminary SAR outcomes have been summarized in Figure 26. The preliminary mechanistic study revealed that these benzazole derivatives appear to have a unique mechanism of action. They were not HDACIs and the HIV-1 transcription was driven neither by the NF-κB signaling pathway nor by stimulating of HIV-1 LTR. These data indicated that benzazole might be a promising chemotype to be developed into potent LRAs with appropriate chemical optimizations and further investigation into the mechanism of actions [128].

#### 2.13.6. Pyridine Derivative

Vitamin B3 niacinamide (**107**) (see Figure 27), a pyridine-derived sirtuin inhibitor, showed notably improved HIV-1 reactivation potency compared to that of the combination of two validated HMTI-based LRAs, chaetocin and BIX01294, in an ex vivo assay for short-term treatment, indicating its clinical potential. Meanwhile, owing to the relatively simple structure, niacinamide might be utilized as a suitable hit to carry out further chemical optimization to find more potent LRAs [129].

#### 2.13.7. Polyphenols

Resveratrol (**108**), as revealed in Figure 28, a natural polyphenol, was capable of reactivating latent HIV-1 without triggering overall T cell activation. Besides, a synergistic HIV reactivation was observed when resveratrol was combined with other conventional LRAs with different mechanisms of action, such as JQ1 (BETI), SAHA (HDACI) and prostratin (PKC activator). A preliminary mechanistic study revealed the latency-reversal activity of resveratrol was due to the activation of HSF1 and increased histone acetylation, but not the activation of silent information regulator 1 (SIRT1), which belongs to a member of the sirtuin family. However, due to relatively low bioavailability, more polyphenol analogues of resveratrol (**109**~**114**) were successively submitted to the HIV-1 reactivation evaluation. The result was that among these polyphenols, only triacetyl resveratrol (**109**) gave comparable latency-reversal potency with the prototype resveratrol, while none of the others could successfully reactivate the latent proviruses in vitro, suggesting more effort is still required to find more potent polyphenol-based LRAs by expanding the chemical diversity of a synthetically or naturally available stilbenoid chemotype. Additionally, the cotreatment with triacetyl resveratrol (**109**) with other LRAs also generated a synergistic effect [130]. Q**205** (**115**), a synthetic resveratrol analogue, was proved to effectively reactivate latent HIV-1 in vitro without inducing of damaging cytokines. A preliminary mechanistic study revealed that the latency-reversal potency of Q205 was attributable to the activation of P-TEFb and promotion of Tat-mediated HIB-1 transcription and binding of RNAPII to the HIV-1 5′-LTR promoter. The results evinced that identification of promising LRAs from naturally available polyphenols and/or chemically optimized resveratrol derivatives might be a feasible and practicable alternative [131].

## 3. LRA Combinations

Since multiple regulatory pathways have participated in the establishment and maintenance of latent reservoirs, a single LRA is obviously not sufficient to accomplish the global viral reactivation. Thus, the combined use of LRAs with different mechanisms might be a more effective means, by affecting various subsets of signaling regulatory pathways, lowering the dose of each component and reducing the unwanted side effects. Given the fact that the combination of PKC activators and HDACIs has proved to be more effective than any LRAs, the PKC activator/HDACI combinational protocols were by far the most reported.

Doria M. and co-workers proved that combination of the PKC activator prostratin with HDACIs would attenuate HDACI toxicity, while the best result was obtained when prostratin was combined with HIDAI romidepsin, by not only stimulating reactivation of latent HIV-1 but also enhancing NKG2D-mediated viral suppression by NK cells [132]. Largazole (SDL 148, **116**), as evinced in Figure 29, a macrocyclic class I-selective HDACI, was validated to be a potent LRA with low toxicity by remodelling chromatin at the HIV-1 5′-LTR promoter. Given the fact that the combination of PKC activators and HDACIs generally gave more effective potency than any one LRA component, the effects of the combination of HDACI largazole and two PKC modulators, bryostatin-1 analogues SUW133 (**117**) and SUW124 (**118**), were determined. As expected, combination of largazole with either of the bryostatin-1 analogues provided remarkable latency-reversal efficacy and induced enhanced levels of proviral expression without triggering overall T cell activation and abnormal cytokine release, indicating that the combination is a potential therapeutic remedy for preclinical advancement [133]. Similarly, the co-administration of the PKC activator gnidimacrin (GM, **63**) with class I-selective HDACI thiophenyl benzamide (TPB, **119**), not only gave 2~3-fold HIV-1 reactivation efficacy but decreased the risk of new viral infection, compared to GM alone at non-toxic concentrations [134]. Additionally, the combination of PKC modulator (−)-indolactam V (**120**) with a pan-HDACI, hydroxamate-tethered phenylbutyrate AR-42 (**121**), resulted in a strong synergistic effect in reversing latent viruses, followed by a significant CD4 down-regulation [135].

In addition to the well-acknowledged PCK activator/HDACI combinations, other combinational LRAs were also reported. For instance, gliotoxin (GTX, **122**), discovered from a high-throughput screen of fungal metabolites, was validated to be an attractive LRA without clear cytotoxicity by disrupting 7SK small nuclear ribonucleoprotein (7SK snRNP) to release P-TEFb. Moreover, cotreatments with GTX with HDACI SAHA or BRM-associated factors (BAF) inhibitor caffeic acid phenetyl esther (**123**) resulted in an unparalleled synergy in reversing HIV-1 latency [136]. Admittedly, combination remedy is not limited to the above-mentioned components, and more and more promising combinations will be identified as the LRA-related research intensifies.

## 4. “Block-and-Lock” Strategy

Despite promising results of various LRAs with different mechanisms in vitro and ex vivo, in vivo and clinical studies revealed that latent reservoirs cannot be successfully cleared. None of the LRAs discovered to date are capable of reactivating all latent HIV viruses in infected host cells while causing minimal side effects, indicating that HIV-1 reactivation induced by using just LRAs, even with a combination of LRAs with the expectation of targeting different reservoirs, is apparently insufficient to accomplish the “shock” element of the “shock-and-kill” regimen. With further in-depth research on the intrinsic survival and maintenance mechanisms of latent reservoirs, a new “block-and-lock” strategy is proposed alternatively, aiming to use latency-promoting agents (LPAs) to prevent low-level or sporadic transcription of integrated proviruses to realize permanent silencing of viruses and an ultimate functional cure of AIDS. Unlike the “shock-and-kill” strategy that requires reactivation of latent proviruses or elimination of latent infected cells, the “block-and-lock” strategy locks viruses in a latent state permanently; thus, they can never be transcribed even for AIDS patients who have stopped routine antiretroviral treatment. The strategy shows great prospect by exhibiting fewer side effects and less impact on the quality of life of patients, making it possible to accelerate a functional cure of AIDS [137,138].

Currently, “block and lock” research is mainly focused on two aspects. One is to find new targets for short hairpin RNAs (shRNA) by screening suitable small interfering RNAs (siRNAs) that target the NF-κB motif located in 5′-LTR of virus, then to find candidates with stronger gene silencing abilities [139]. Although siRNA-based drugs are promising, achieving life-long silence to a highly mutated virus that can infect multiple cell types remains a tremendous challenge. Firstly, the effectiveness of various siRNA delivery systems needs to be further confirmed in animal models, and the delivery efficiency also needs to be improved. Secondly, explosive replication of resistant mutant strains should be avoided during application of siRNA. Finally, siRNA-based drugs are expensive, and meanwhile it is infeasible to employ intravenous administration every few days, making it difficult to maintain an effective concentration of siRNA in HIV-1 infected patients for a few years. Therefore, the siRNA-tethered therapy is difficult to popularize extensively, especially in developing and underdeveloped countries. Additional technical breakthroughs, including but not limited to increasing chemical stability, improving pharmacokinetics and delivery efficiency, are still needed before it is widely used in clinical practice. However, for individuals who have been resistant to available antiviral drugs, siRNA-based agents might be optimal alternatives [140].

As described above, there are still numerous difficulties to be dealt with when artificial shRNAs are introduced into HIV-1 infected cells through a suitable delivery system. Therefore, the other attempt to accomplish the “block and lock” strategy is using small molecular LPAs. For example, Zhu et al. found that levosimendan (**124**, see Figure 30), a vasodilator and calcium sensitizer used for heart failure, is a promising LPA by directly screening FDA-approved compound libraries. Levosimendan could inhibit acute viral replication in initial CD4^+^ T cells and prevent reactivation of latent HIV-1 proviruses [141]. Given that levosimendan has been approved for treating acute heart failure, it is more feasible to be used in clinical promotions as compared to other LPAs.

Nonetheless, compared to the well-validated “shock-and-kill” strategy, there is still a long ways to go for the alternative “block and lock” protocol, since there is evidence that the “block and lock” strategy may worsen the immune system in immunodeficient patients and that siRNA-mediated transcriptional silencing can be restrained by Tat protein [142].

## 5. Summary and Outlook

The “Shock and kill” strategy will still be the mainstream in the future research on latent reservoirs. However, there are still many difficulties in the development of potent LRAs to postpone the “shock” component. Firstly, there are arguments about the survival and maintenance mechanisms of latent reservoirs, and currently there is still no uniform approach for evaluating the size of latent reservoirs. Hence, deciphering these mechanisms is pivotal in designing more efficient approaches to eliminate latent HIV-1 infection. Secondly, the lack of a suitable in vivo evaluation model is another bottleneck that hinders the quick identification of more effective LRAs for clinical advancement, since most of in vivo latently infected models used to evaluate novel compounds are actually SIV-infected macaque models treated with HAART. Although the SIV virus has a high homology with human HIV-1, and symptoms of SIV-infected macaques are very similar to those of HIV-1 infected humans, there are still many differences in genome sequence between the two viruses. Thirdly, the HIV-1 provirus lurks in different cells which have distinct phenotypes and metabolic characteristics. Affected by a client’s medication history, the latency mechanism may vary from different patients, or vary in different cells from the same patient. Simultaneously, the number and distribution of HIV-1 proviruses in different latently infected cells may change over time; so does the molecular mechanism of latency, enabling latency to be a dynamic process; for that reason, a particular LRA can only be effective for a specific period of time. Finally but not the least, although a variety of novel LRAs have favorable in vitro activities by increasing transcription of viruses in varied latent cell models, there is minimal in vivo HIV-1 reactivation effectiveness in most cases. Therefore, eradicating latent reservoirs in AIDS patients remains a tremendous challenge. To achieve a functional cure of AIDS, future work should consider measures for more specifically modifying viral transcription to identify more effective LRAs by targeting accurately targets, or by using a reasonable LRA combination remedy.

## Figures and Tables

**Figure 1 molecules-28-00003-f001:**
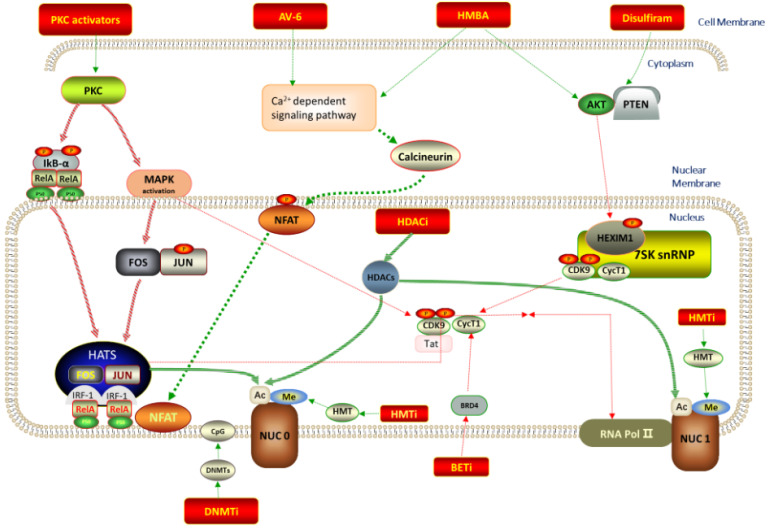
Schematic overview of molecular mechanisms of latent viral reservoirs and corresponding LRAs.

**Figure 2 molecules-28-00003-f002:**
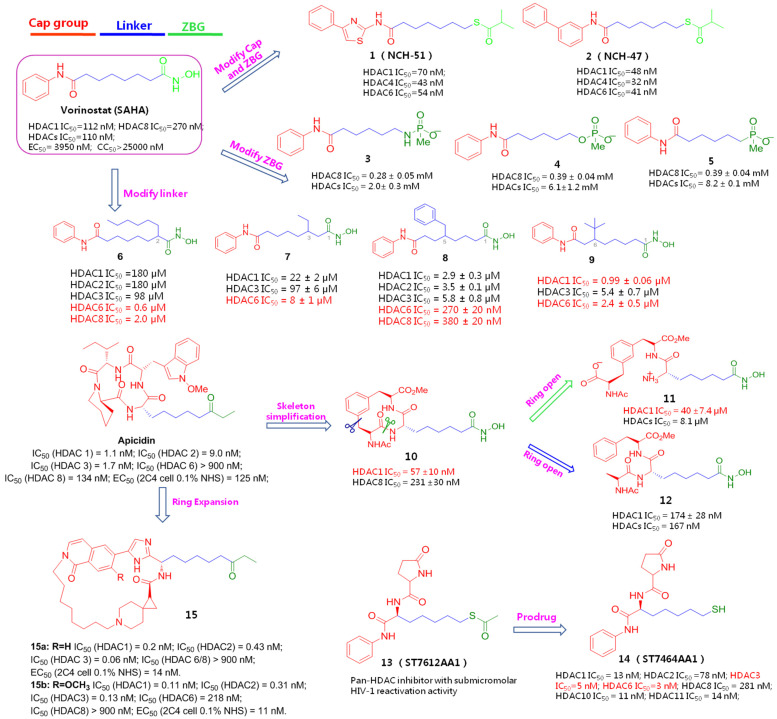
Chemical structures, HDAC inhibitory potencies and latency-reversing activities of a spectrum of HDACIs derived from SAHA or Apicidin.

**Figure 3 molecules-28-00003-f003:**
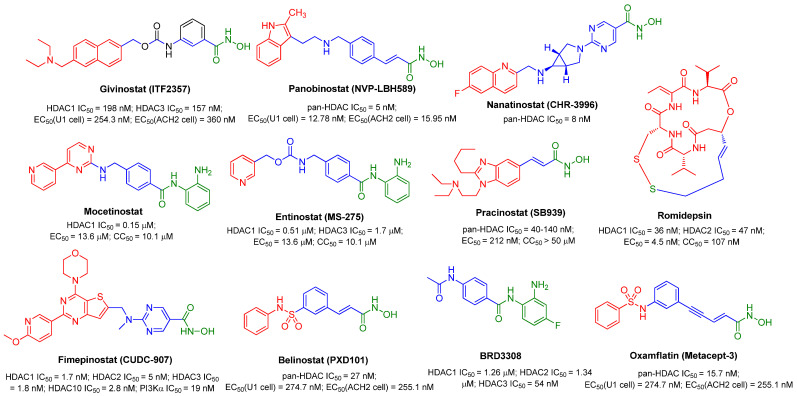
Several pan-HDACIs or class I-selective HDACIs that have entered various clinical trials for anticancer purposes were also submitted to the anti-latency evaluations.

**Figure 4 molecules-28-00003-f004:**
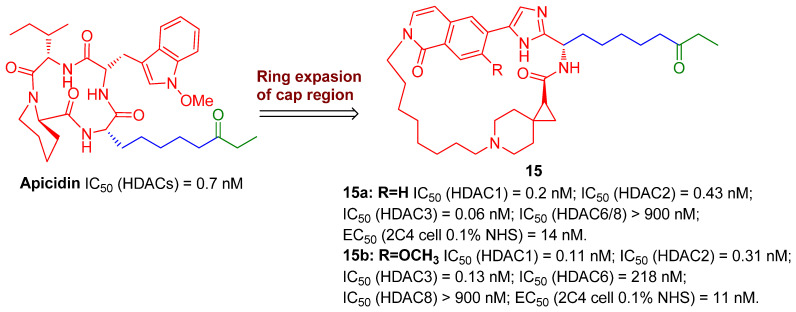
Class I-selective macrocyclic HDACIs 15 developed from Apicidin.

**Figure 5 molecules-28-00003-f005:**
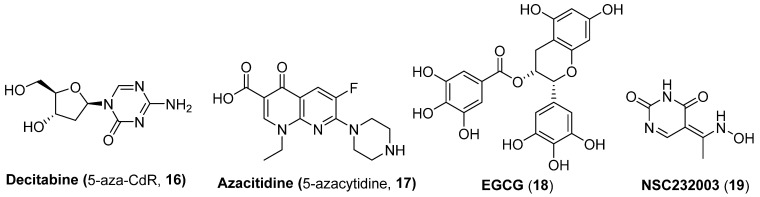
Chemical structures of DMTIs (**16** and **17**) and UHRF1s (**18** and **19**) with latency-reversal potencies.

**Figure 6 molecules-28-00003-f006:**
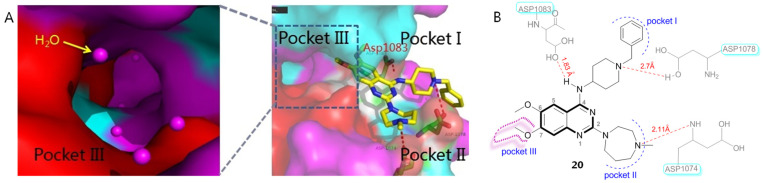
(**A**) Crystal structure of three active binding domains in HMT (PDB code: 3K5K) and (**B**) binding mode of BIX01294 (20) with HMT.

**Figure 7 molecules-28-00003-f007:**
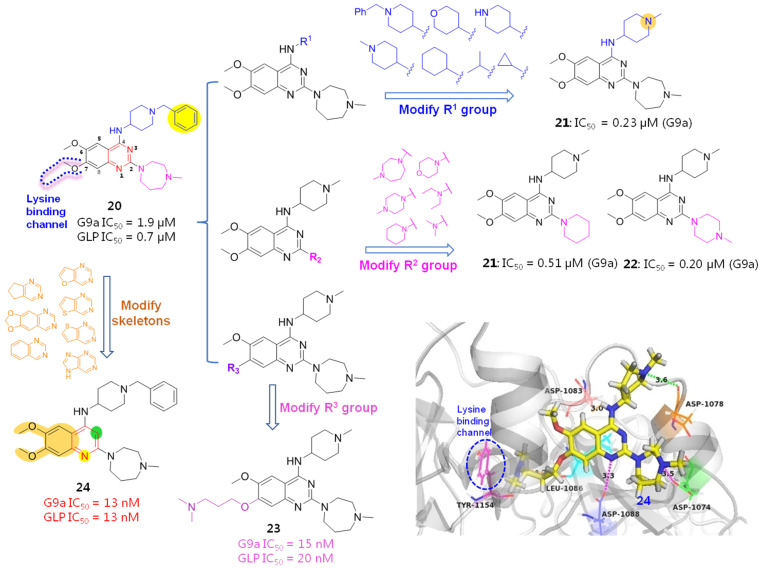
Structure-based modifications based on BIX01294 (**20**) led to more potent HMTIs,** 24** and **25**, and the binding pattern of **24** with G9a was also presented.

**Figure 8 molecules-28-00003-f008:**
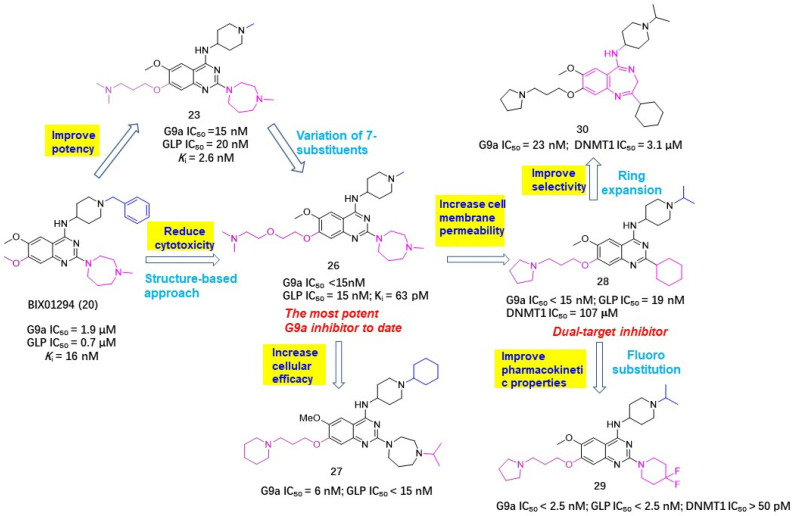
Structure-based modifications based on BIX01294 (**20**) led to several more potent HMTIs with improved drug-like profiles.

**Figure 9 molecules-28-00003-f009:**
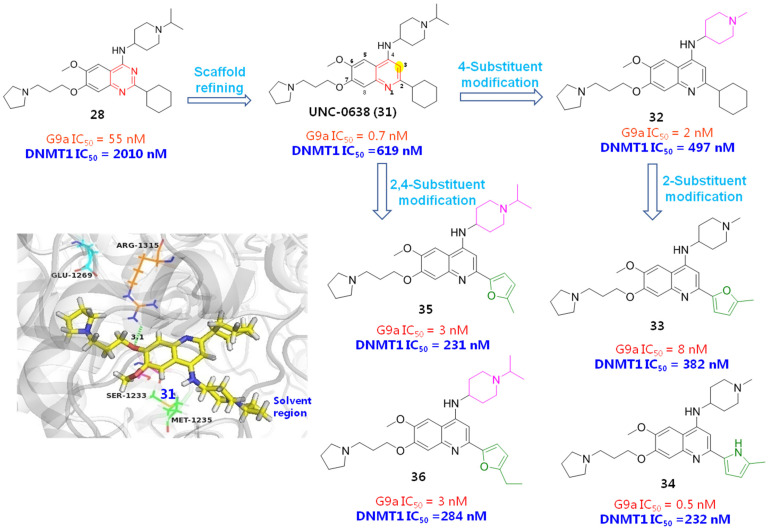
Structural optimizations based on G9a/DNMT1 dual inhibitor (**28**) by refining the quinazoline scaffold and substituents at 2- and 4-positions led to quinoline-based derivatives **31**~**36** with more potent G9a/DNMT1 inhibitory activities; binding pattern of 31 with mouse DNMT1 was also displayed to guide further optimizations.

**Figure 10 molecules-28-00003-f010:**
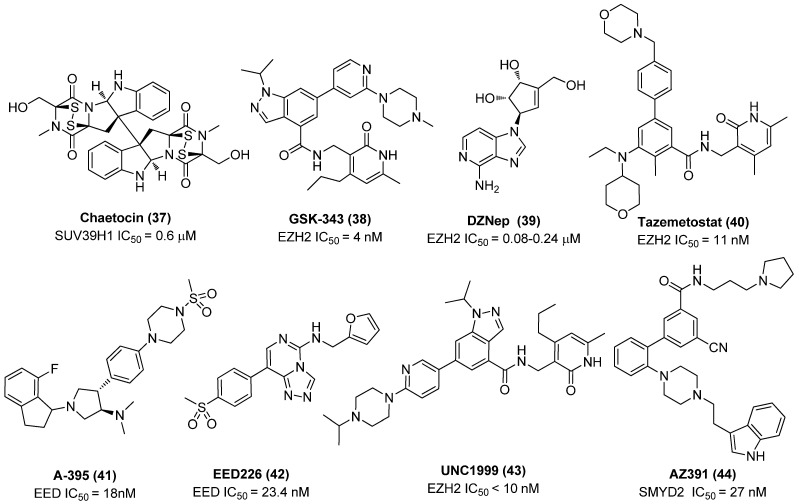
Chemical structures of several identified HMTIs that possess HIV-1 latent reversal effects.

**Figure 11 molecules-28-00003-f011:**
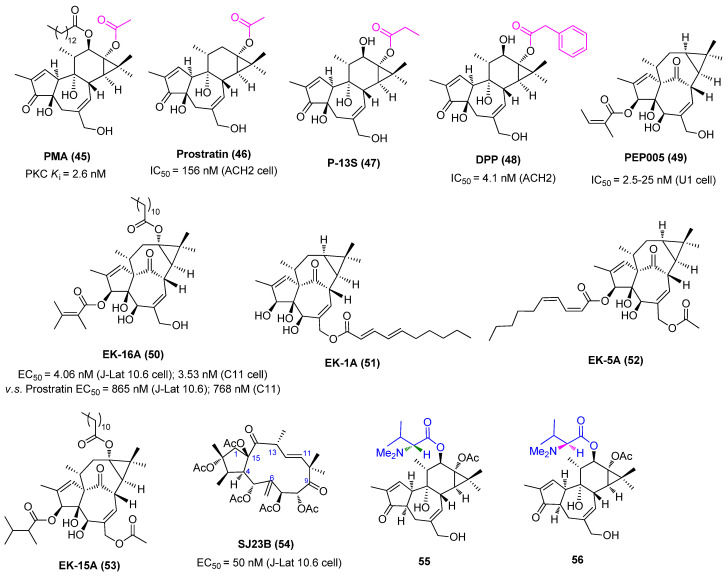
Representative PKC activators (**45**~**64**) with potent latency-reversal effects.

**Figure 12 molecules-28-00003-f012:**
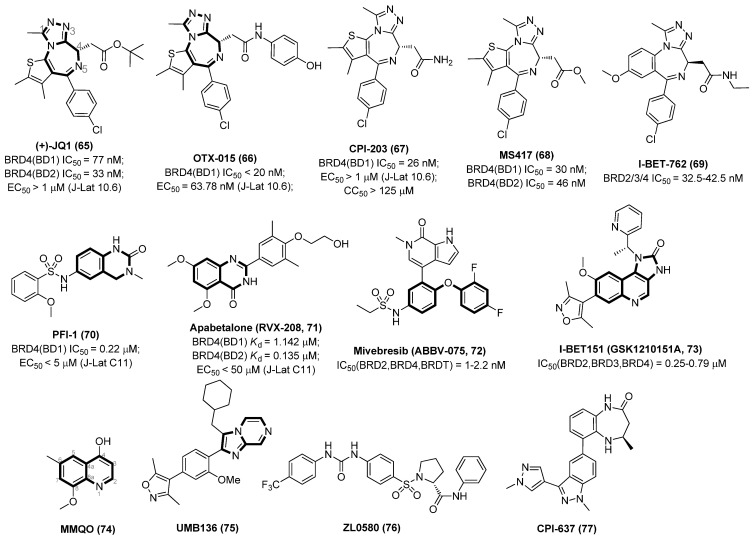
Chemical structures of representative BETIs as LRAs (**65**~**77**).

**Figure 13 molecules-28-00003-f013:**
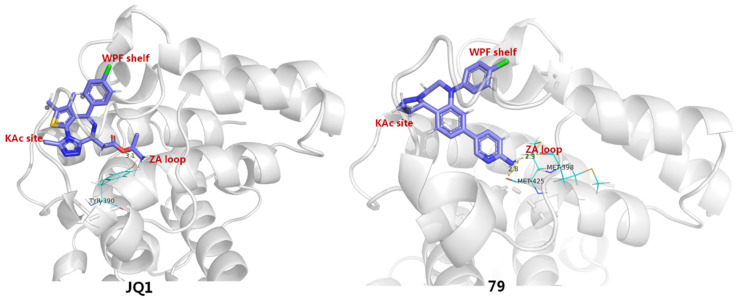
Binding mode comparison between compounds JQ1 and 79.

**Figure 14 molecules-28-00003-f014:**
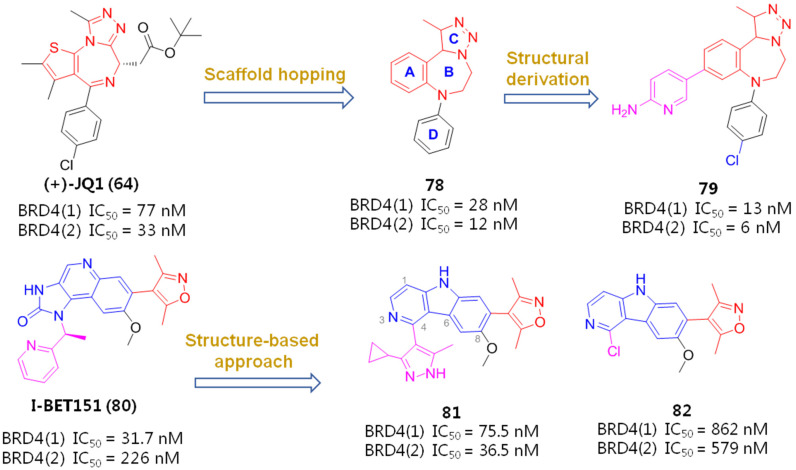
Chemical modifications from JQ1 and I-BET151 led to several more potent BETIs.

**Figure 15 molecules-28-00003-f015:**
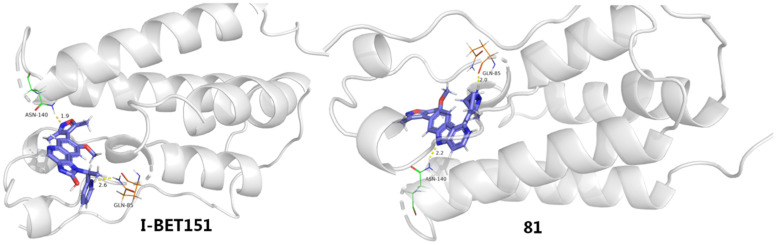
Binding mode comparison between I-BET151 and **81**.

**Figure 16 molecules-28-00003-f016:**
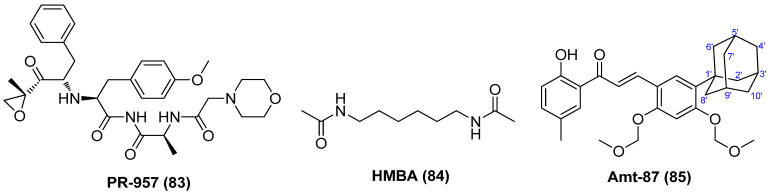
Chemical structures of P-TEFb activators (**83**~**85**) with HIV-1 reactivation efficacies.

**Figure 17 molecules-28-00003-f017:**
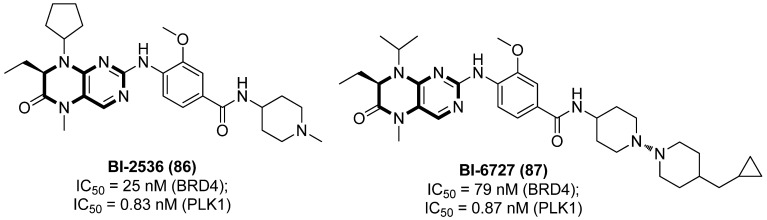
Pteridine-based PLK1/BET dual inhibitors **86** and **87** as LRAs.

**Figure 18 molecules-28-00003-f018:**
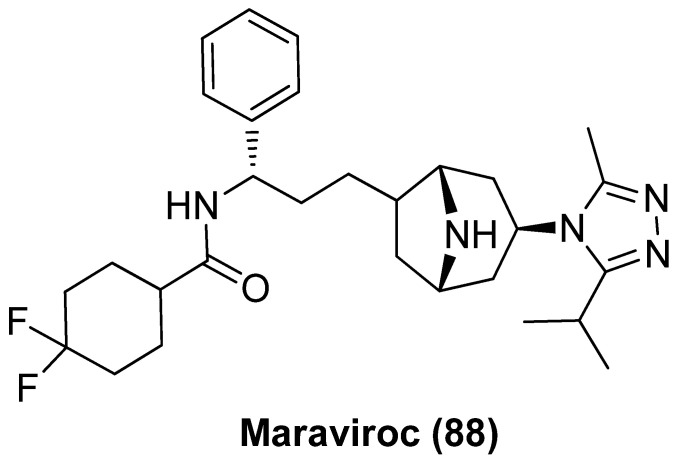
CCR5 antagonist maraviroc (**88**) as LRA.

**Figure 19 molecules-28-00003-f019:**
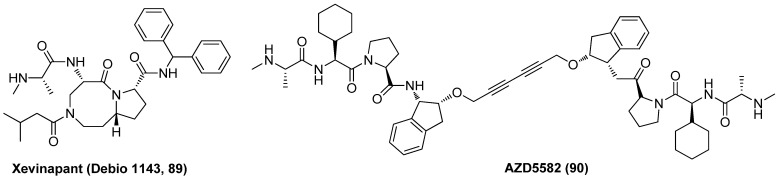
Chemical structure of two IAP antagonists **89** and **90** as LRAs.

**Figure 20 molecules-28-00003-f020:**
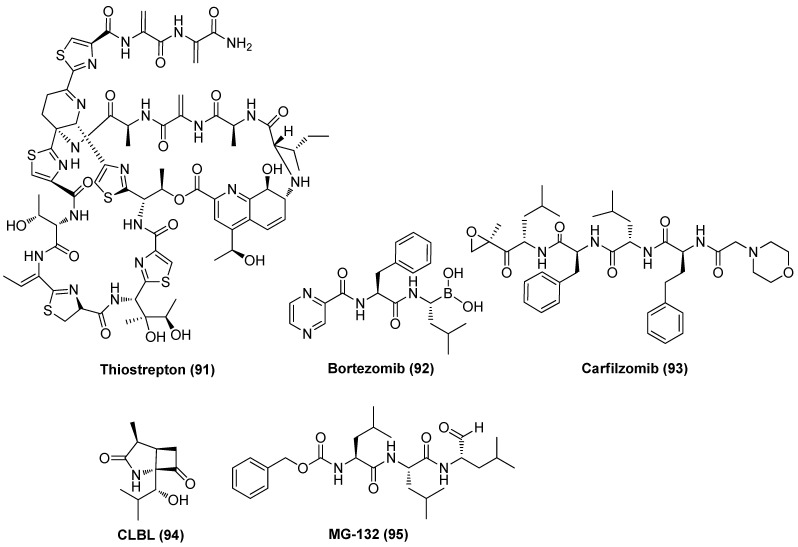
Chemical structures of proteasome inhibitors **91**~**95** as LRAs.

**Figure 21 molecules-28-00003-f021:**
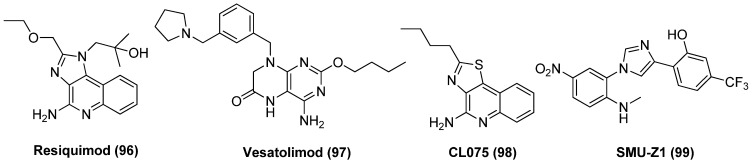
Chemical structures of TLR agonists **96**~**99** as LRAs.

**Figure 22 molecules-28-00003-f022:**
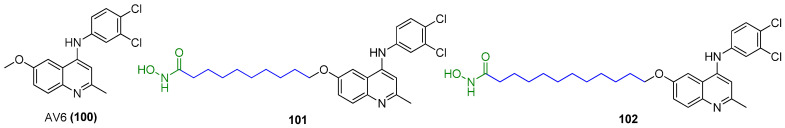
Chemical structures of quinoline derivatives **100**~**102** as LRAs.

**Figure 23 molecules-28-00003-f023:**
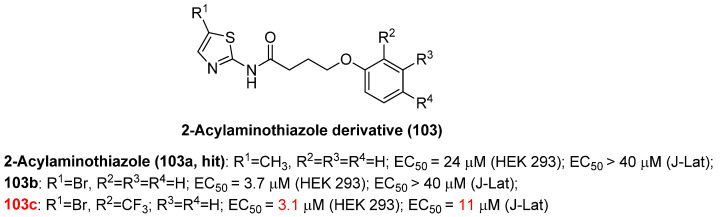
Chemical structures of 2-acylaminothiazole derivatives **103**a–c.

**Figure 24 molecules-28-00003-f024:**
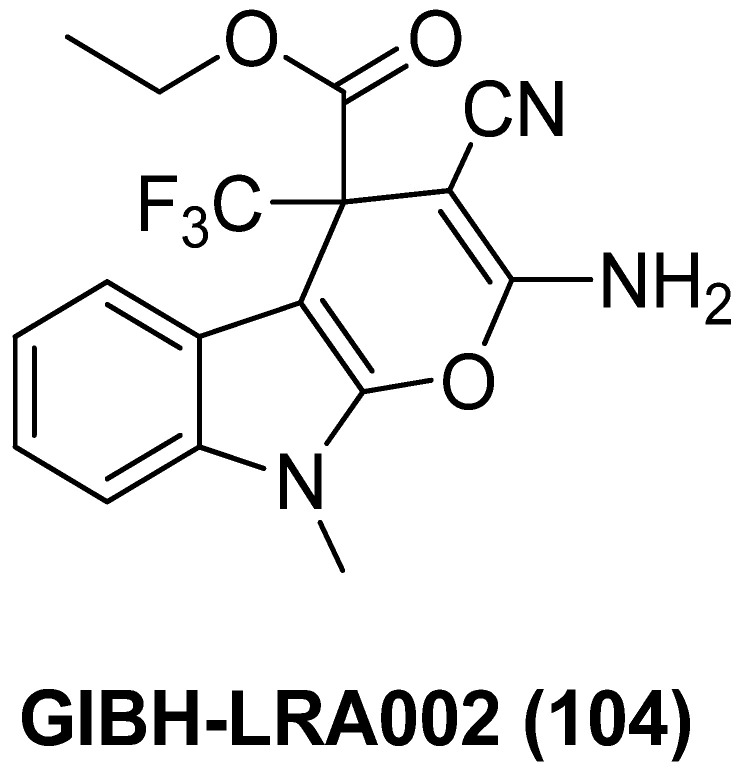
Chemical structure of pyranoindole derivative GIBH-LRA002 (**104**).

**Figure 25 molecules-28-00003-f025:**
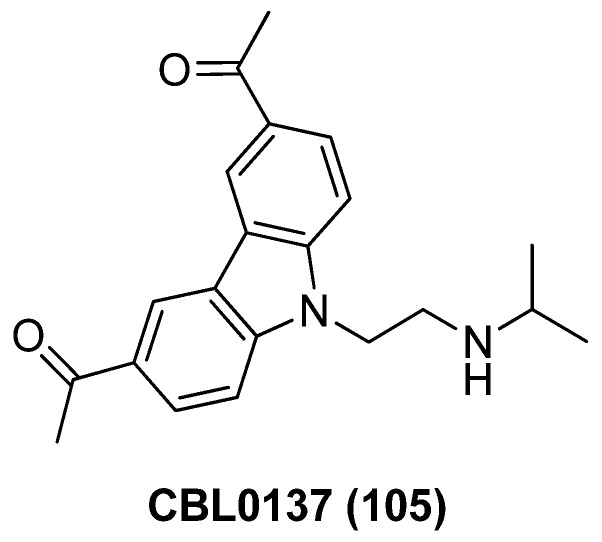
Chemical structure of carbazole derivative CBL0137 (**105**).

**Figure 26 molecules-28-00003-f026:**
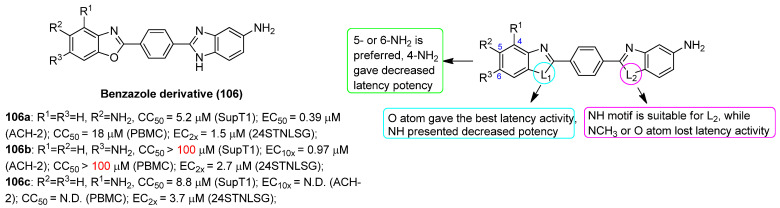
Chemical structures and preliminary SAR summary of benzazole derivative **106**.

**Figure 27 molecules-28-00003-f027:**
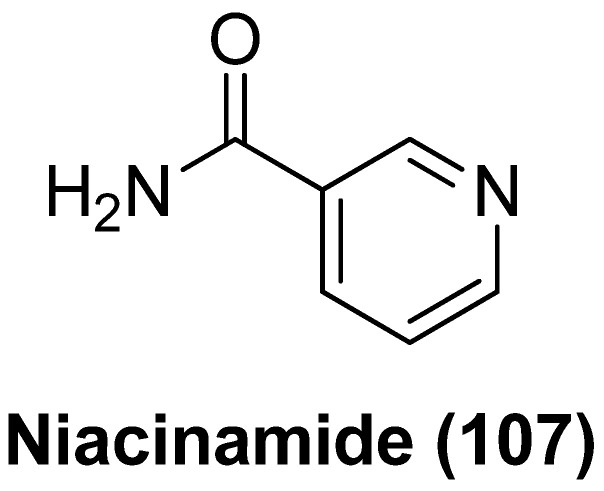
Chemical structure of niacinamide (**107**).

**Figure 28 molecules-28-00003-f028:**
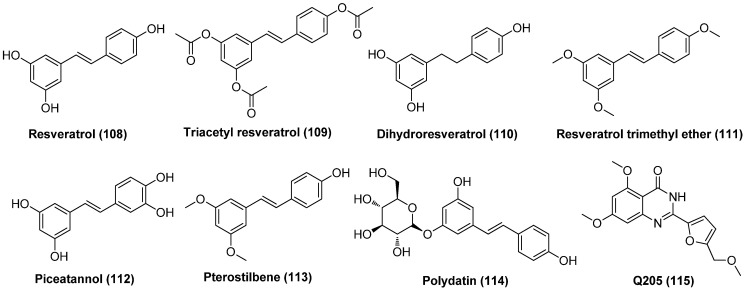
The chemical structure of several polyphenols. Resveratrol (**108**) and its analogue, triacetyl resveratrol (**109**), displayed clear HIV-1 reactivation potencies.

**Figure 29 molecules-28-00003-f029:**
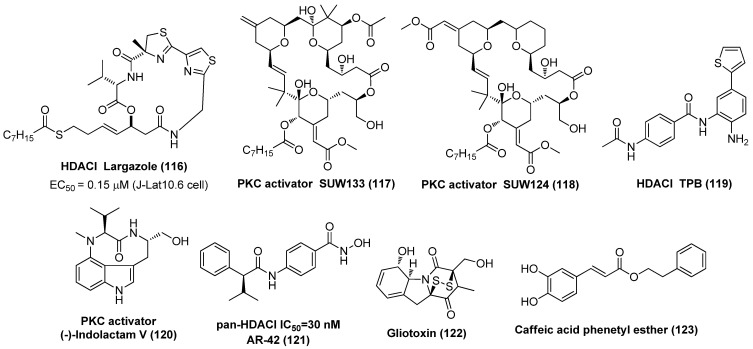
LRAs used as combinational components.

**Figure 30 molecules-28-00003-f030:**
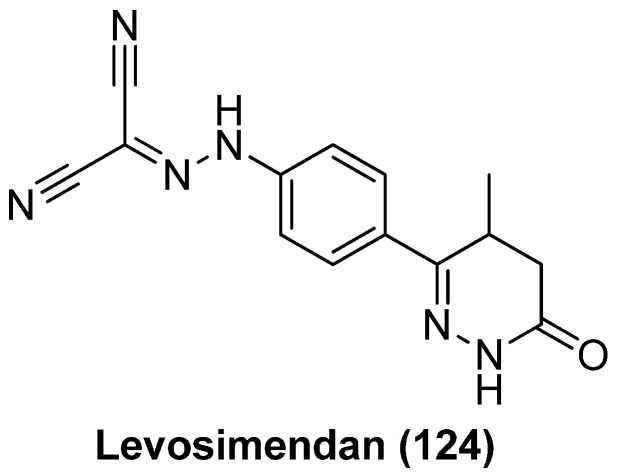
Levosimendan (**124**) is a LPA identified from FDA-approved compound libraries.

## Data Availability

Not applicable.

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
