# Peer review of "Medicinal Chemistry of Anti-HIV-1 Latency Chemotherapeutics: Biotargets, Binding Modes and Structure-Activity Relationship Investigation"

_molecules, 2022, doi:10.3390/molecules28010003_

Round 1

Reviewer 1 Report

Manuscript ID: molecules-2098750

Title: Medicinal Chemistry of Anti-HIV-1 Latency Chemotherapeutics: Biotargets, Binding Modes and Structure-Activity Relationship Investigation

The manuscript authored by Yan-Kai Wang, Long Wei, Wei Hu, Peri-Xia Yu, Hai-Peng Yu and Xun Li reported Latency Chemotherapeutics, including biotargets, binding modes and structure-Activity relationship

The manuscript is well written, with some typical grammar mistakes; the information is sufficient, well summarized and includes the most relevant information on the topic, so it the suitable for publication, subject to comments below in the scope of molecules.

Attached pdf file with specific comments

These are some specific questions and observations

1.- The figures that include the molecules with their kinetic parameters (2, 3, 4, 7, 8, 9, 10, 11, 12, 14, 17, 26), it is recommended to separate the parameters and list them in a table for a better understanding.

The information described in each subtopic of “Possible Biotargets and related LRAs” is valuable; a brief conclusion with further explanation would complement these sections.

Author Response

1.- The figures that include the molecules with their kinetic parameters (2, 3, 4, 7, 8, 9, 10, 11, 12, 14, 17, 26), it is recommended to separate the parameters and list them in a table for a better understanding.

®Response: We are greatly appreciated of this suggestion. However, we believed it is easy to understand the SAR outcomes by easily finding the chemical structures and corresponding kinetic parameters. Therefore, we would better to remain the original figure unchanged.

  1. The information described in each subtopic of “Possible Biotargets and related LRAs” is valuable; a brief conclusion with further explanation would complement these sections.

®Response: We are greatly appreciated of this suggestion. After proof-reading all the manuscript, brief conclusions for each biotarget and corresponding LRAs have been given, particular the most studied biotargets at present, such as HDAC, PKC and BET. For example, for the conclusion of PKC activators, a brief conclusion has been added as follows.

“In brief, PKC activators are still attractive LRAs which are particularly effective in combination with other LRAs with different mechanisms.”

Reviewer 2 Report

The present manuscript deals well with the issues it has taken up and is an excellent collection of data from various aspects on HIV latency and the development of latency-reducing inhibitors. It can be considered  for publication after additional information on the following issues:

1. SAR analysis of DMTIs be included for a better understanding of the activities of these molecules.

2. Docking studies on all types of drug targets be included.

3. Some minor grammatical errors be eliminated.

Author Response

  1. SAR analysis of DMTIs be included for a better understanding of the activities of these molecules.

®Response: We are greatly appreciated of this suggestion. SAR analysis of DMTIs has been revised and included.

  1. Docking studies on all types of drug targets be included.

®Response: We are greatly appreciated of this suggestion. In authors’ opinion, it is unnecessary to present all the docking results of all types of LRAs. For one reason, not all LRAs based on varied biotargets gave excellent latency reversal activities, and as is often the case, only LRA with promising potency is worth doing the docking experiments so as to find binding information to guide further modifications. For another reason, not all biotargets have resolved x-ray crystallographic structure in PDB. Besides, the added docking studies will make the article too long.

  1. Some minor grammatical errors be eliminated.

®Response: We are greatly appreciated of this suggestion. With the assistance of reviewers and editor, the grammatical errors have been checked and revised accordingly.